# MMSearch:
# Unveiling the Potential of Large Models as Multi-modal Search Engines

**Dongzhi Jiang**[1*], **Renrui Zhang**[1,3*†], **Ziyu Guo**[2], **Yanmin Wu**[5], **Jiayi Lei**[4]
**Pengshuo Qiu**[4], **Pan Lu**[6], **Zehui Chen**[3], **Guanglu Song**[7]
**Peng Gao**[4], **Yu Liu**[7], **Chunyuan Li**[3], **Hongsheng Li**[1,8‡]

[1]CUHK MMLab & [2]MiuLar Lab    [3]ByteDance    [4]Shanghai AI Laboratory
[5]Peking University    [6]Stanford University    [7]Sensetime Research    [8] CPII under InnoHK
{dzjiang,renruizhang,ziyuguo}@link.cuhk.edu.hk

 * Equal contribution    † Project lead    ‡ Corresponding author

## Abstract

The advent of Large Language Models (LLMs) has paved the way for AI search engines, e.g., SearchGPT, showcasing a new paradigm in human-internet interaction. However, most current AI search engines are limited to text-only settings, neglecting the multimodal user queries and the text-image interleaved nature of website information. Recently, Large Multimodal Models (LMMs) have made impressive strides. Yet, whether they can function as AI search engines remains under-explored, leaving the potential of LMMs in multimodal search an open question. To this end, we first design a delicate pipeline, **MMSearch-Engine**, to empower any LMMs with multimodal search capabilities. On top of this, we introduce **MMSearch**, a comprehensive evaluation benchmark to assess the multimodal search performance of LMMs. The curated dataset contains 300 manually collected instances spanning 14 subfields, which involves no overlap with the current LMMs' training data, ensuring the correct answer can only be obtained within searching. By using MMSearch-Engine, the LMMs are evaluated by performing three individual tasks (requery, rerank, and summarization), and one challenging end-to-end task with a complete searching process. We conduct extensive experiments on closed-source and open-source LMMs. Among all tested models, GPT-4o with MMSearch-Engine achieves the best results, which surpasses the commercial product, Perplexity Pro, in the end-to-end task, demonstrating the effectiveness of our proposed pipeline. We further present error analysis to unveil current LMMs still struggle to fully grasp the multimodal search tasks, and conduct ablation study to indicate the potential of scaling test-time computation for AI search engine. We hope MMSearch may provide unique insights to guide the future development of multimodal AI search engine. Project Page is at https://mmsearch.github.io.

## 1 Introduction

Search engines (Brin & Page, 1998) have been the main tools for humans to navigate through the overwhelming quantity of online resources. Recently, Large Language Models (LLMs) (OpenAI, 2023a;b; Touvron et al., 2023a) have demonstrated impressive performance on various zero-shot downstream applications. On top of this, AI search engine (OpenAI, 2024c), which integrates LLMs with traditional search engines, stands among one of the most promising ones. It points the direction of the next-generation interaction paradigm of human and Internet. Combining the language

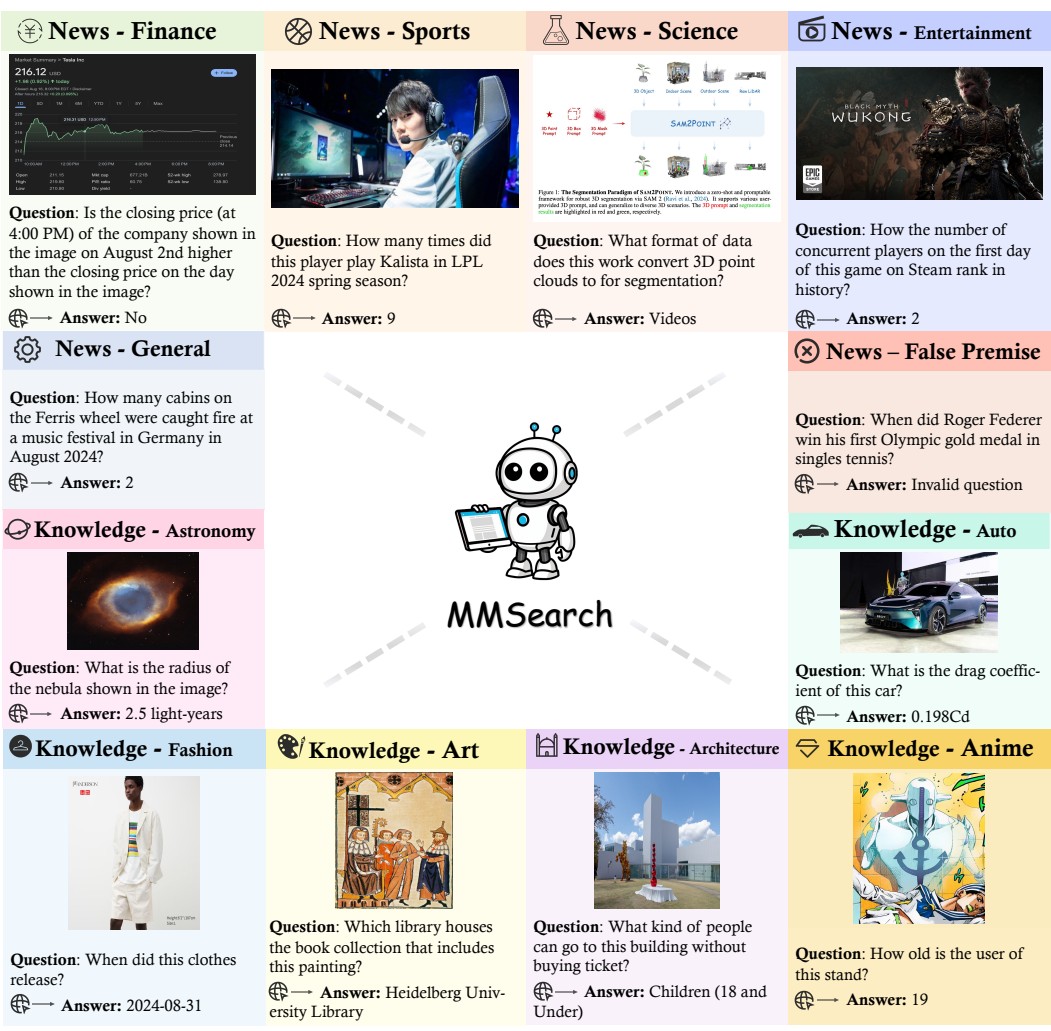

Figure 1: **Overview of the MMSEARCH Benchmark.** MMSEARCH aims to evaluate any LMM's potential to be a multimodal AI search engine. The benchmark contains two primary areas: latest news and rare knowledge to ensure no overlap with LMM's inherent knowledge.

understanding ability of LLMs and up-to-date information from the Internet, AI search engines could better grasp the user's intention and summarize contextual-aligned answers from the raw web information. These systems can only process textual queries and interpret textual web content, significantly constraining user query scenarios and information-seeking methods (Barbany et al., 2024; Xie et al., 2024). This limitation impacts both the range of input queries and the accuracy of results (Jiang et al., 2024a; Chen et al., 2021; Lù et al., 2024), particularly given the complexity and interleaved nature of modern websites (Liu et al., 2024c). For example, consider a scenario where you possess numerous medals belonging to your grandfather but are unaware of their specific names. A multimodal AI search engine could match photographs of these medals with an interleaved table of images and text retrieved from the Internet, thereby identifying each medal. In contrast, text-only search engines can neither take photographs for searching nor understand the interleaved table. Hence, a multimodal AI search engine is crucial for advancing information retrieval and analysis.

On the other hand, with the recent rapid advancements, Large Multimodal Models (LMMs) (Liu et al., 2023a; Lin et al., 2023; OpenAI, 2023c; Gao et al., 2024; Zhang et al., 2024b) have showcased significant abilities across diverse scenarios, including general image understanding (Fu et al., 2023; Liu et al., 2023b; Yu et al., 2023), expert image reasoning (Zhang et al., 2024d; Gao et al., 2023a; Zhang et al., 2024c; Guo et al., 2025b), multi-image perception (Li et al., 2024a; Wang et al., 2024; Jiang et al., 2024b; Li et al., 2024c), and spatial environment perception (Guo et al., 2023; Yang

et al., 2023; Han et al., 2023). Despite these developments, a framework for LMMs to function as multimodal AI search engines remains largely unexplored. Consequently, the potential of LMMs in multimodal searching also remains a significant open question.

To bridge this gap, we first present MMSEARCH-ENGINE, a multimodal AI search engine pipeline, empowering any LMMs with advanced search capabilities. MMSEARCH-ENGINE maximizes the utilization of LMMs' multimodal information comprehension abilities, incorporating both visual and textual website content as information sources. On top of this, we introduce MMSEARCH, a multimodal AI search engine benchmark to comprehensively evaluate LMMs' searching performance. The design of MMSEARCH-ENGINE facilitates the zero-shot evaluation of any LMMs within the context of AI search engine. Our experiment covers state-of-the-art closed-source (OpenAI, 2023c; Anthropic, 2024; Gemini Team, 2023) and open-source LMMs (Li et al., 2024b; Qwen Team, 2024; Chen et al., 2024d; Ye et al., 2024). Our efforts are summarized as follows:

i. **MMSEARCH-ENGINE, a multimodal AI search engine pipeline for LMMs,** empowering large models for multimodal searching. In contrast with the conventional text-only AI search engines, MMSEARCH-ENGINE fully integrates multimodal information in two ways: (i) for queries containing images, we conduct web searches across both textual and visual modalities. We utilize Google Lens (len) to identify critical visual information from the input image; (ii) all search results are presented in both textual and visual formats, ensuring a comprehensive understanding of the interleaved website content. The working flow of MMSEARCH-ENGINE contains multi-round interaction between LMM and the Internet. The LMM needs to first *requery* the user question into a search-engine-friendly format. Then, the LMM *reranks* the retrieved websites based on its helpfulness. Finally, the LMM is required to *summarize* the answer based on the most informative webpage content selected from the rerank. Thanks to the design of the pipeline, we propose a step-wise evaluation strategy on the three core tasks within the searching process: *requery*, *rerank*, and *summarization*. The final score is weighted by the end-to-end evaluation results and scores of the three core tasks.

ii. **MMSEARCH, a comprehensive benchmark for multimodal AI search engines,** which, to our best knowledge, serves as the first evaluation dataset to measure LMMs' multimodal searching capabilities. Our benchmark categorizes searching queries into two primary areas: *News* and *Knowledge*, as shown in Fig. 1. We employed different strategies for these two areas to ensure the challenging nature of the benchmark. *News* area covers the latest news at the time of data collection (August, 2024). This is to guarantee the answers to the queries will not be present in the training data of LMMs. As for the area of *Knowledge*, we collect queries requiring rare knowledge and then select the queries unable to be answered by current SoTA LMMs such as GPT-4o (OpenAI, 2024b) or Claude-3.5 (Anthropic, 2024). The two areas sum up to 14 subfields. In total, MMSEARCH encompasses 300 meticulously collected queries, with 2901 unique images.

iii. ***Extensive experiments and error analysis for future development direction.*** We evaluate popular closed-source models and open-source LMMs on MMSEARCH. GPT-4o achieves the best overall performance across different tasks. Surprisingly, our MMSEARCH-ENGINE equipeed with SoTA LMMs, such as GPT-4o and Claude 3.5 Sonnet, even surpasses the prominent commercial AI search engine Perplexity Pro (Perplexity) in the end-to-end task. Our thorough error analysis reveals that current LMMs still struggle to generalize to multimodal search-specific tasks. Their poor requery and rerank capabilities significantly limit their ability to correctly identify useful websites and extract relevant answers. Additionally, we identify five error types for requery and summarization tasks, respectively. We find that current LMMs cannot fully understand the requery task and do not know how to query the search engine. As for the summarization task, LMMs often have difficulty in extracting useful information, either from text or images. These capabilities are essential for LMMs to function as robust multimodal search engines and require further development. We also conduct a preliminary ablation study to explore the potential of scaling test-time computation versus scaling model size (OpenAI, 2024a). Initial results indicate that scaling test-time computation demonstrates superior performance in this task.

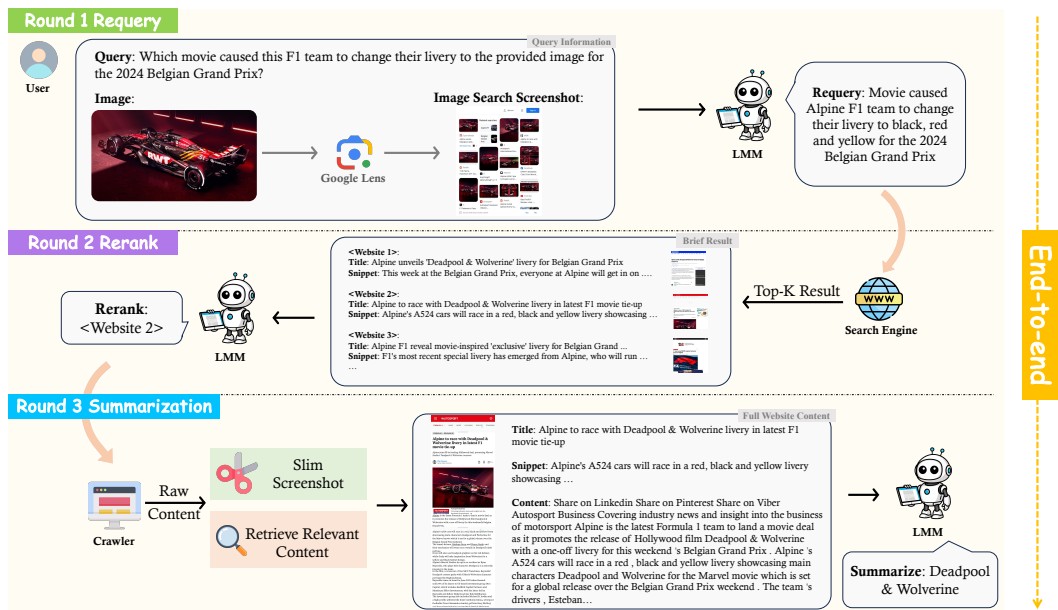

Figure 2: **The Pipeline of MMSEARCH-ENGINE.** The process comprises three sequential stages executed by a LMM: (i) requery, (ii) rerank, and (iii) summarization. In the end-to-end evaluation task, the LMM completes these three stages sequentially to generate the final output.

## 2 MMSEARCH

In Section 2.1, we first detail the design of our multimodal AI search engine pipeline, which serves as both data collection and evaluation tools. Then, in Section 2.2, we detail the data composition and collection of the curated multimodal search benchmark MMSEARCH. Then, in Section 2.3, we elaborate on our step-wise evaluation strategy. Finally, we detail the dynamic nature of our benchmark in Section 2.4.

### 2.1 MMSEARCH-ENGINE: A MULTIMODAL AI SEARCH ENGINE PIPELINE

The searching process is a complex action including multi-round interactions between LMMs and conventional search engines. We develop a delicate pipeline that queries LMMs multiple times to accomplish this task. Leveraging the image comprehension capabilities of LMMs, we incorporate two types of visual data. First, we incorporate Google Lens (len) to search for information from the image. The second type of visual data is the screenshot of the retrieved websites, in the purpose of preserving the original format of website content. Our framework is shown in Fig. 2. Below we detail how an LMM works with this pipeline, which comprises three sequential phases:

i. *Requery.* The query direct from users may contain references to certain information in the image, e.g., the *News-Finance* example shown in Fig. 1. Since a conventional search engine only accept text-only input, it is necessary for LMM to translate the image content and combine it with the query to ask a valid question to it. In addition, the raw user query may be ambiguous or inefficient sometimes (Chan et al., 2024; Ma et al., 2023), reformulating the query to be more clear is also a must for LMM. If the user query contains an image, we incorporate the screenshot of the image search result from the google lens (len). We treat the user query, user image, and the image search screenshot as basic information of the query. This information will be input to LMM in every round in the pipeline. For the requery round, we prompt LMM to output a requery to a conventional search engine.

ii. *Rerank.* The requery is sent to a search engine API, e.g., DuckDuckGo, to retrieve top $K$ relevant websites. Depending on the requery quality, not all retrieved websites are necessarily relevant for query answering. Hence, we prompt LMM to select one most informative website for answer summarization. Due to the LMM's context length limitations and the

Table 1: **Key Statistics of MMSEARCH.**

| Statistic | Number |
|---|---|
| Total questions | 300 |
| - Questions with images | 171 (57.0%) |
| - Questions without images | 129 (43.0%) |
| Total Websites | 2,280 |
| Total Areas/Subfields | 2/14 |
| Number of unique images | 2,901 |
| - Query images | 163 |
| - Google search images | 163 |
| - Top section screenshot images | 2,280 |
| - Full-page screenshot images | 295 |
| Number of unique questions | 289 |
| Number of unique requeries | 289 |
| Number of unique reranked websites | 2,400 |
| Number of unique answers | 264 |
| Maximum question length | 41 |
| Maximum answer length | 12 |
| Average question length | 14.0 |
| Average answer length | 1.9 |

Figure 3: **Area and Subfield Distribution of MMSEARCH.**

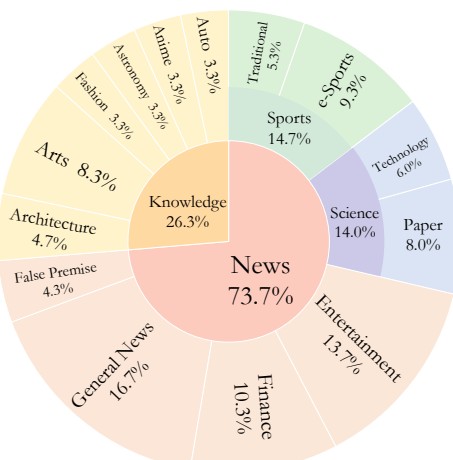

extensive content of websites, we provide only essential information of each website, which we term *brief results*. These brief results include the title, the snippet, and a screenshot of the webpage's top section, which serves as the input for LMM's reranking. The inclusion of the screenshot serves two purposes. First, the screenshot offers a visual cue to assess the web's credibility, as a well-organized website often appears more trustworthy than one cluttered with advertisements (Fogg et al., 2001; Sillence et al., 2004). Additionally, the screenshot may contain essential visual information. For instance, it might include images similar or identical to query images, as shown in the *Website 2* in Fig. 2.

iii. *Summarization.* We start by crawling the selected website to gather all the available information. We parse the HTML to obtain the raw textual content and capture a full-page screenshot of the website. However, there are two issues: the raw content tends to be extensively lengthy and disorganized, while substantial areas in the full-page screenshot are blank due to the ad blocks on the website. These two issues lead to a large number of input tokens filled with irrelevant information. To enhance data efficiency, we slim the screenshot and retrieve the relevant content before inputting them to LMM. For the full-page screenshot, we identify the blank areas and remove them iteratively, detailed in Appendix F. As for the text content, we apply a text embedding model (Chen et al., 2024a) to retrieve a maximum of 2K tokens relevant to the requery from the raw content. We define the slimmed screenshot and the retrieved content as *full website content*. Finally, we input the full website content, website title, and website snippet, along with the query information, to LMM for summarizing the answer.

## 2.2 DATA COMPOSITION AND COLLECTION

To thoroughly assess multimodal search proficiency, we compile a comprehensive problem set covering a broad spectrum of news topics, specialized knowledge domains, and query image patterns. This widespread collection for MMSEARCH aims to simulate diverse user searching scenarios, ensuring a robust evaluation of LMMs' capabilities in multimodal search.

**Data Composition and Categorization.** Our benchmark aims to isolate LMMs' inherent knowledge and assess their actual search capabilities. We focus on two primary areas: *News* and *Knowledge*. For the *News* area, the queries are related to the latest news at the time of data collection (August 2024). This guarantees no overlap between the current LMMs' training data and questions in our benchmark. All questions in this area are recorded with their occurrence time. For fairness, LMMs with recently updated knowledge should be tested on queries that occurred after their latest data update. Due to its time-sensitive nature, the *News* area serves as a dynamic part of our benchmark. Please refer to Section 2.4 for details. As for the *Knowledge* area, we focus on rare knowledge in targeted domains. Each question proposed by an annotator is verified to be beyond the

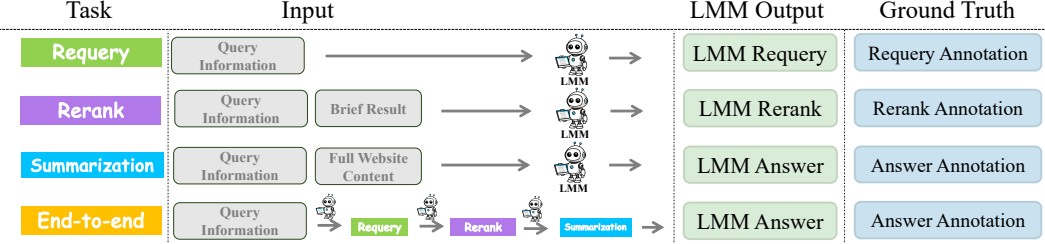

Figure 4: **Outline of Evaluation Tasks, Inputs, and Outputs.** Our evaluation contains four tasks. The requery, rerank, and summarization tasks assess the LMM's proficiency in individual pipeline rounds. The end-to-end task simulates a real-world search scenario by sequentially executing all three stages. An example of the input and output is shown in Fig. 10 in the supplementary material.

capabilities of state-of-the-art Large Language Models (LLMs) such as GPT-4o (OpenAI, 2024b) or Claude 3.5 Sonnet (Anthropic, 2024). The *Knowledge* area serves as a static component of our benchmark and remains constant over time. We collect a total of 300 queries across the 2 primary areas and 14 subfields. This dataset size balances comprehensive evaluation with efficiency, considering the multi-round interactions with the search engine and Internet for each query. Detailed statistics for data composition and categorization are presented in Table 1 and Fig. 3. Definitions of each subfield are in Appendix G.2.

**Data Collection and Review Process.** Thanks to the design of our pipeline, the data collection process follows similar procedure introduced in the pipeline. An annotator is first required to propose a query and provide its answer, either sourced from the latest news or rare knowledge. The annotator then formulates a requery based on the query information. After $K$ websites are retrieved from the search engine, the annotator is required to divide all $K$ websites into three sets based on the brief results: *valid* (likely to contain the answer), *unsure* (relevance is difficult to determine), and *invalid* (entirely irrelevant to the question). We mandate that at least one website must be classified as *valid*; if this criterion is not met, the annotator is required to adjust the requery to obtain new search results. Finally, we randomly pick one website from the *valid* set and obtain its full content. To ensure the question is answerable, another annotator is employed to give an answer to the query based on the full content. If the answer is incorrect, the question needs to be revised until it is answerable.

## 2.3 EVALUATION PROTOCOL

In contrast with previous LMM benchmarks, the multimodal search process of LMM contains multiple rounds. Only the end-to-end evaluation of the final answer is inadequate to reveal the models' deficiency in each core searching step. For example, the errors made by the model may occur during the summarization process, but it might also stem from choosing an incorrect website during the reranking stage. To this end, we propose a step-wise strategy to evaluate the LMMs' capability on the three core searching steps, in addition to the end-to-end evaluation.

- **End-to-end score ($\mathbf{S}_{e2e}$):** We compute the F1 score between the predicted answer and the ground truth to judge if the answer is correct.
- **Requery score ($\mathbf{S}_{req}$):** We apply the average of ROUGE-L and BLEU-1 scores to measure the similarity between the model's requery and human-annotated requery.
- **Rerank score ($\mathbf{S}_{rer}$):** The rerank score is derived from the LMM's selection among $K$ pre-defined websites. The score values is 1.0 for valid set, 0.5 for unsure set, and 0 for invalid set or incorrect format.
- **Summarization score ($\mathbf{S}_{sum}$):** Again, we compute the F1 score of LMM's answer based on a pre-defined website content against ground truth.

The input, output, and ground truth of the four tasks are visualized in Fig. 4. The final score is weighted by these four scores. We assign the highest weight (75%) to the end-to-end task, as it reflects the real-world multimodal search capability. The remaining 25% is distributed among the intermediate steps: 10% each for the rerank and summarization tasks, and 5% for the requery task. The lower weight for the requery task accounts for the inherent uncertainty in this process. The scoring process can be formulated as:

$$\mathbf{S}_{final} = 0.75 \cdot \mathbf{S}_{e2e} + 0.05 \cdot \mathbf{S}_{req} + 0.1 \cdot \mathbf{S}_{rer} + 0.1 \cdot \mathbf{S}_{sum} \quad (1)$$

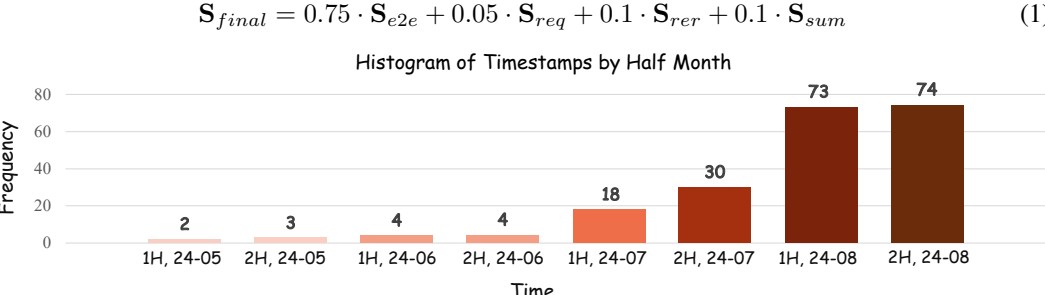

Figure 5: **Timestamps Distribution of Questions in the *News* Area.** All events of our collected data occurred after May 2024. The majority of data concentrates on August. This ensures the data captures only recent events, falling beyond the knowledge cutoff dates of LMMs. The False Premise subfield is not included, since it is infeasible to determine the timestamp for an event that never occurred.

## 2.4 BENCHMARK EVOLUTION

In Fig. 5, we showcase the statistics of data timestamp distribution in the *News* area. Our dataset spans from 1st May 2024 to 31th August 2024. By the time of evaluation, we inspect the knowledge cutoff dates of the closed-source models. Claude 3.5 Sonnet reports a knowledge cutoff of April 2024, while both GPT-4V and GPT-4o state they lack information from 2024. For open-source models, we examine their release dates and training data, confirming that none possess knowledge beyond May 2024. This temporal gap ensures the fairness of our evaluation, as the models' performance solely reflects their multimodal search capabilities rather than pre-existing knowledge. We will update the *News* area if a new LMM's training data may overlap with our collection period.

## 3 EXPERIMENT

In this section, we conduct a systematic evaluation of existing LMMs on MMSEARCH. We first introduce the experimental setup in Section 3.1. Then, we detail the quantitative results in Section 3.2 and narrate the error analysis in Section 3.3. Finally, we conduct an in-depth ablation study of scaling test-time compute versus scaling model size in Section E.1 in the supplementary material.

## 3.1 EXPERIMENT SETUP

**Evaluation Models** We examine the performance of foundation models across three distinct categories on MMSEARCH: (a) *Commercial AI Search Engines*, represented by Perplexity (Perplexity). We test the pro version of Perplexity, which takes only the user query and image as input. Since SearchGPT (OpenAI, 2024c) has not been public yet, we do not test on it. (b) *Closed-source LMMs*, represented by models like GPT-4V (OpenAI, 2023c), GPT-4o (OpenAI, 2024b), and Claude 3.5 Sonnet (Anthropic, 2024), and (c) *Open-source LMMs*, featuring models such as LLaVA-OneVision-7B (Li et al., 2024b) (Qwen2-7B (Yang et al., 2024a)), LLaVA-OneVision-72B (Li et al., 2024b) (Qwen2-72B (Yang et al., 2024a)), LLaVA-NeXT-Interleave (Li et al., 2024c) (Qwen1.5-7B (Yang et al., 2024a)), InternVL2 (Chen et al., 2024d) (InternLM2.5-7B-Chat (Cai et al., 2024)), InternLM-XC2.5 (Zhang et al., 2024a) (InternLM2-7B (Cai et al., 2024)), Qwen2-VL-7B (Qwen Team, 2024) (Qwen2-7B (Yang et al., 2024a)), Qwen2-VL-72B (Qwen Team, 2024) (Qwen2-72B (Yang et al., 2024a)), mPlug-Owl3 (Ye et al., 2024) (Qwen2-7B (Yang et al., 2024a)), Idefics3 (Laurençon et al., 2024) (LLaMA3.1-7B-Instruct (AI@Meta, 2024)), and Mantis (Jiang et al., 2024b) (LLaMA3-7B (AI@Meta, 2024)). Note that the open-source LMMs' sizes are 7B unless otherwise specified.

**Implementation Details.** We set the number of retrieved websites $K$ as 8. We include two image input resolution settings. For the default settings, the longest edge of the input image is resized to match the largest resolution of the vision encoder of LMM. This ensures the image not to be cropped to multiple images and will only take up the minimum of tokens for image input. For any resolution settings, we input the image without resizing. More details are available in Section F.

Table 2: **Evaluation Results of Four Tasks in MMSEARCH.** We report the scores of news and knowledge areas and their average score in each task. Subscript *AnyRes* indicates original resolution image input; otherwise, low-resolution images were used. The highest scores for closed-source and open-source LMMs are marked in red and blue. *For open-source LMMs, we adopt the models with 7B parameters unless otherwise specified.*

| Model | All | | | End-to-end | | | Requery | | | Rerank | | | Summarize | | |
|---|---|---|---|---|---|---|---|---|---|---|---|---|---|---|---|
| | Avg | News | Know. | Avg | News | Know. | Avg | News | Know. | Avg | News | Know. | Avg | News | Know. |
| *Baselines* | | | | | | | | | | | | | | | |
| Human | 69.2 | 69.6 | 68.1 | 68.2 | 68.6 | 67.1 | 43.7 | 45.0 | 40.1 | 85.7 | 87.3 | 81.2 | 72.8 | 71.4 | 76.7 |
| *Commercial AI Search Engines* | | | | | | | | | | | | | | | |
| Perplexity Pro (Perplexity) | - | - | - | 47.8 | 52.7 | 34.1 | - | - | - | - | - | - | - | - | - |
| *Closed-source LMMs  with MMSEARCH-ENGINE* | | | | | | | | | | | | | | | |
| Claude 3.5 Sonnet (Anthropic, 2024) | 53.5 | 53.1 | 54.7 | 49.9 | 49.3 | 51.6 | 42.0 | 43.6 | 37.7 | 80.2 | 78.7 | 84.2 | 59.4 | 60.3 | 57.0 |
| GPT-4V (OpenAI, 2023c) | 55.0 | 55.0 | 55.3 | 52.1 | 52.2 | 51.9 | 45.7 | 49.2 | 35.8 | 79.3 | 76.9 | 86.1 | 57.4 | 56.7 | 59.4 |
| GPT-4o (OpenAI, 2024b) | 62.3 | 61.2 | 65.3 | 60.4 | 59.0 | 64.5 | 46.8 | 49.9 | 38.0 | 83.0 | 82.4 | 84.8 | 63.1 | 62.2 | 65.6 |
| *Open-source LMMs  with MMSEARCH-ENGINE* | | | | | | | | | | | | | | | |
| Mantis (Jiang et al., 2024b) | 18.7 | 19.8 | 15.9 | 15.8 | 16.4 | 14.3 | 20.1 | 24.6 | 7.4 | 39.7 | 41.0 | 36.1 | 19.2 | 22.0 | 11.5 |
| InternLM-XC2.5 (Zhang et al., 2024a) | 22.2 | 22.8 | 20.5 | 22.9 | 23.6 | 20.8 | 25.0 | 24.3 | 27.0 | 0.0 | 0.0 | 0.0 | 37.7 | 38.6 | 35.0 |
| InternLM-XC2.5$_{AnyRes}$ | 22.3 | 23.9 | 17.5 | 23.2 | 25.4 | 16.9 | 21.7 | 19.8 | 26.9 | 0.0 | 0.0 | 0.0 | 37.7 | 38.6 | 35.1 |
| LLaVA-NeXT-Interleave (Li et al., 2024c) | 28.3 | 29.2 | 25.6 | 23.0 | 23.8 | 20.5 | 26.2 | 30.7 | 13.5 | 55.3 | 58.6 | 46.2 | 42.5 | 40.0 | 49.3 |
| mPlug-Owl3 (Ye et al., 2024) | 32.1 | 34.8 | 24.4 | 24.6 | 28.1 | 14.9 | 32.6 | 36.7 | 21.2 | 74.3 | 73.5 | 76.6 | 45.6 | 45.6 | 45.4 |
| mPlug-Owl3$_{AnyRes}$ | 33.9 | 35.5 | 29.3 | 27.3 | 29.4 | 21.2 | 31.8 | 36.1 | 19.9 | 74.5 | 72.9 | 79.1 | 43.9 | 43.6 | 44.6 |
| InternVL2 (Chen et al., 2024d) | 34.3 | 35.7 | 30.2 | 30.9 | 32.5 | 26.2 | 32.3 | 36.1 | 21.6 | 46.5 | 49.3 | 38.6 | 48.5 | 45.9 | 55.8 |
| InternVL2$_{AnyRes}$ | 34.1 | 34.2 | 33.7 | 30.0 | 30.3 | 29.0 | 31.4 | 35.5 | 19.7 | 53.2 | 52.3 | 55.7 | 46.9 | 44.4 | 53.7 |
| Idefics3 (Laurençon et al., 2024) | 36.2 | 38.5 | 29.6 | 29.3 | 32.3 | 20.8 | 31.0 | 37.3 | 13.5 | 76.5 | 73.3 | 85.4 | 50.3 | 51.1 | 48.1 |
| Idefics3$_{AnyRes}$ | 35.7 | 38.2 | 28.7 | 30.1 | 32.9 | 22.3 | 27.2 | 32.2 | 13.2 | 72.7 | 73.1 | 71.5 | 45.2 | 46.3 | 42.1 |
| LLaVA-OneVision (Li et al., 2024b) | 36.6 | 39.4 | 28.9 | 29.6 | 33.1 | 19.7 | 35.8 | 40.3 | 23.2 | 72.8 | 73.5 | 70.9 | 53.5 | 51.8 | 58.5 |
| Qwen2-VL$_{AnyRes}$ (Qwen Team, 2024) | 45.3 | 44.1 | 48.7 | 40.3 | 39.2 | 43.5 | 39.0 | 41.9 | 30.8 | 76.7 | 73.8 | 84.8 | 54.7 | 52.7 | 60.4 |
| LLaVA-OneVision (72B) | 50.1 | 50.1 | 50.2 | 44.9 | 45.1 | 44.1 | 42.9 | 45.9 | 34.3 | 82.2 | 80.5 | 86.7 | 61.4 | 59.1 | 67.7 |
| Qwen2-VL$_{AnyRes}$ (72B) | 52.7 | 52.0 | 54.5 | 49.1 | 48.8 | 49.8 | 44.7 | 47.2 | 37.6 | 76.7 | 72.9 | 87.3 | 59.6 | 57.5 | 65.7 |

## 3.2 EXPERIMENTAL ANALYSIS

To thoroughly investigate the multimodal searching capabilities, we present the evaluation results of different models on MMSEARCH following the proposed step-wise evaluation strategy in Table 2 and fourteen subfields in Table 3. We now provide a detailed discussion of notable findings and their implications for multimodal search capabilities.

**Any-resolution input only provides slight or no improvement.** Of the tested LMMs, four models, which are InternLM-XC2.5, InternVL2, mPlug-Owl3, and Idefic3, all support both low-resolution (LowRes) and any-resolution input (AnyRes). As one would expect, AnyRes input enables better OCR and perception of the image. However, we only observe slight or even no enhancement comparing the difference between the LowRes performance and its AnyRes counterpart. Take mPlug-Owl3 as an example, AnyRes input surpasses LowRes input on overall score by 1.8%, end-to-end score by 2.7%, and rerank on 0.2%. While it falls behind LowRes on requery by 0.8% and summarization by 1.7%. This suggests that the OCR and perception quality do not bottleneck the search performance. Rather, the suboptimal performance appears to stem from the LMMs' inherent lack of robust search capabilities.

**Current LMMs still have significant shortcomings in requery and rerank.** Comparing the average score of the end-to-end task with that of the summarization task, we find that the summarization score consistently surpasses the end-to-end task by a large margin, both in the closed-source and open-source models. The minimum margin is 2.7% for GPT-4o, while the maximum is 23.9% for LLaVA-OneVision-7B. This discrepancy can be attributed to the differences in the tasks' input quality. While the summarization task input always contains the answer, the end-to-end task's third-round input quality depends on the model's requery and rerank quality in previous rounds. The magnitude of this performance gap reflects the disparity between a model's summarization ability and its capacity for requery and rerank tasks. The larger the difference, the larger the capability gap. Observing the result, we find that this gap of most open-sourced models exceeds 14%, while the closed-sourced models are all below 10%. This suggests all current LMMs needs improvement

Table 3: **Evaluation Results on Different Subfields in MMSEARCH.** SPO: Traditional Sports; ESP: E-Sports; ENT: Entertainment; GEN: General News; PAP: Paper; TEC: Technology; FIN: Finance; FAL: False Premise; ART: Arts; ARC: Architecture; AST: Astronomy; ANI: Anime; AUT: Auto; FAS: Fashion. The highest scores for closed-source and open-source LMMs are marked in red and blue. *For open-source LMMs, we adopt the models with 7B sizes unless otherwise specified.*

| Model | All | News | | | | | | | | | Knowledge | | | | | | |
|---|---|---|---|---|---|---|---|---|---|---|---|---|---|---|---|---|---|
| | | Avg | SPO | ESP | ENT | GEN | PAP | TEC | FIN | FAL | Avg | ART | ARC | AST | ANI | AUT | FAS |
| *Closed-source LMMs  with MMSEARCH-ENGINE* | | | | | | | | | | | | | | | | | |
| Claude 3.5 Sonnet (Anthropic, 2024) | 53.5 | 53.0 | 37.4 | 50.2 | 63.6 | 49.2 | 52.8 | 67.7 | 43.4 | 63.1 | 54.7 | 59.0 | 42.9 | 70.9 | 56.5 | 60.6 | 36.4 |
| GPT-4V (OpenAI, 2023c) | 55.0 | 54.9 | 49.3 | 48.5 | 67.4 | 43.5 | 37.3 | 64.2 | 59.1 | 90.2 | 55.3 | 54.6 | 45.6 | 65.1 | 63.0 | 52.2 | 56.2 |
| GPT-4o (OpenAI, 2024b) | 62.3 | 61.2 | 63.7 | 61.2 | 72.3 | 51.3 | 48.6 | 68.6 | 60.0 | 76.8 | 65.3 | 73.8 | 52.0 | 57.6 | 76.8 | 68.4 | 55.8 |
| *Open-source LMMs  with MMSEARCH-ENGINE* | | | | | | | | | | | | | | | | | |
| InternLM-XC2.5 (Zhang et al., 2024a) | 22.2 | 22.8 | 22.6 | 13.6 | 28.9 | 23.5 | 15.5 | 27.8 | 32.9 | 3.1 | 20.5 | 29.2 | 24.0 | 20.8 | 2.0 | 20.4 | 11.8 |
| InternLM-XC2.5_AnyRes | 22.2 | 23.9 | 25.2 | 12.8 | 30.6 | 21.0 | 18.7 | 31.4 | 32.8 | 13.9 | 17.6 | 22.9 | 25.6 | 10.8 | 2.0 | 28.1 | 4.7 |
| Mantis (Jiang et al., 2024b) | 18.8 | 19.8 | 17.4 | 9.7 | 25.3 | 19.6 | 22.1 | 35.3 | 18.2 | 6.2 | 15.9 | 23.0 | 17.5 | 12.5 | 14.8 | 12.7 | 3.3 |
| LLaVA-NeXT-Interleave (Li et al., 2024c) | 28.3 | 29.3 | 23.3 | 18.5 | 41.9 | 33.6 | 26.2 | 27.3 | 28.0 | 14.7 | 25.6 | 31.1 | 22.7 | 49.7 | 16.5 | 19.7 | 7.1 |
| InternVL2 (Chen et al., 2024d) | 34.3 | 35.7 | 38.7 | 20.7 | 46.8 | 33.2 | 24.6 | 44.1 | 44.5 | 27.2 | 30.1 | 41.7 | 33.5 | 26.9 | 15.4 | 36.8 | 7.9 |
| InternVL2_AnyRes | 34.0 | 34.2 | 32.0 | 26.6 | 45.4 | 35.0 | 18.7 | 23.7 | 43.1 | 36.3 | 33.7 | 49.5 | 26.4 | 36.5 | 18.3 | 35.2 | 15.5 |
| mPlug-Owl3 (Ye et al., 2024) | 32.1 | 34.8 | 30.0 | 20.6 | 46.7 | 33.7 | 22.3 | 30.4 | 45.7 | 41.5 | 24.4 | 28.7 | 24.6 | 33.5 | 18.3 | 24.2 | 10.5 |
| mPlug-Owl3_AnyRes | 33.9 | 35.5 | 42.0 | 27.1 | 49.7 | 34.4 | 24.5 | 30.4 | 41.3 | 18.9 | 29.3 | 34.4 | 20.3 | 40.9 | 18.8 | 32.8 | 24.4 |
| Idefics3 (Laurençon et al., 2024) | 36.2 | 38.5 | 43.5 | 24.2 | 50.1 | 41.7 | 27.5 | 36.2 | 38.8 | 37.4 | 29.6 | 34.2 | 32.1 | 32.1 | 29.6 | 32.7 | 9.1 |
| Idefics3_AnyRes | 35.7 | 38.2 | 40.5 | 27.3 | 48.0 | 41.4 | 33.1 | 36.9 | 42.1 | 17.4 | 28.8 | 40.5 | 26.2 | 27.1 | 14.2 | 40.8 | 7.3 |
| LLaVA-OneVision (Li et al., 2024b) | 36.6 | 39.4 | 31.8 | 27.1 | 50.0 | 38.5 | 31.4 | 52.1 | 47.4 | 23.5 | 28.9 | 38.0 | 26.4 | 34.3 | 21.0 | 29.6 | 11.2 |
| Qwen2-VL_AnyRes (Qwen Team, 2024) | 45.3 | 44.1 | 47.8 | 33.9 | 49.4 | 45.8 | 45.6 | 38.1 | 49.8 | 31.1 | 48.7 | 70.0 | 32.7 | 43.5 | 42.5 | 54.4 | 23.3 |
| LLaVA-OneVision (72B) | 50.1 | 52.3 | 53.2 | 45.4 | 62.1 | 45.6 | 41.5 | 64.8 | 47.8 | 37.2 | 44.5 | 63.4 | 42.8 | 52.4 | 24.3 | 70.0 | 31.7 |
| Qwen2-VL_AnyRes (72B) | 52.7 | 52.9 | 50.0 | 46.3 | 66.2 | 36.3 | 55.0 | 57.2 | 56.6 | 52.0 | 52.1 | 58.0 | 45.8 | 65.4 | 45.7 | 68.6 | 41.9 |

of their requery and rerank ability, especially for open-source models. Mantis is one exception of open-source models with a margin of only 3.4%. This means its poor summarization capability bottlenecks its end-to-end performance. Qwen2-VL-72B's 10.5% gap, also falling below 14%, highlights its superiority among other open-source LMMs.

**Closed-source LMMs are better-performed than open-sourced LMMs on overall performance.** For the final score, closed-source LMMs consistently outperform the open-source LMMs. GPT-4o achieves the highest overall score of 62.3%, demonstrating superior zero-shot multimodal search capabilities. While Qwen2-VL-72B leads among open-source models, it still lags behind GPT-4o by 9.6%. The performance gap widens to 11.3% on the most challenging end-to-end task and further expands to 20.1% for 7B open-source LMMs. These significant disparities highlight substantial room for improvement in open-source models.

**SoTA LMMs with our MMSEARCH-ENGINE surpass commercial AI search engines in the end-to-end task.** We also evaluate the pro version of Perplexity (Perplexity), a prominent commercial AI search engine that accepts both image and text queries, on our dataset. Perplexity pro can accept both image and text in the user query. Surprisingly, although Perplexity also leverages SoTA LMMs like GPT-4o and Claude 3.5 Sonnet, it largely underperforms MMSEARCH-ENGINE equipped with the same model in the end-to-end task. Even more remarkably, MMSEARCH-ENGINE can even surpass Perplexity with Qwen2-VL-72B, an open-source LMM. *This suggests that our **MMSEARCH-ENGINE** provides a better open-source plan for multimodal AI search engine.* The performance gap validates MMSEARCH-ENGINE's design effectiveness and highlights the value of testing various LMMs within our pipeline, since the pipeline can indeed achieve remarkable performance when using powerful LMMs. Upon investigating Perplexity's sub-optimal performance, we discovered that it appears to utilize only a rudimentary image search algorithm, if any. This limitation leads to poor identification of the key objects in the image and failure to retrieve relevant information. Our findings underscore the effectiveness of MMSEARCH-ENGINE's design, particularly the incorporation of a robust image search step, which plays a crucial role in accurately recognizing important information from the input image.

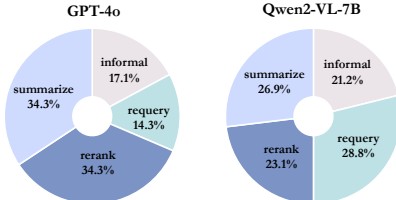
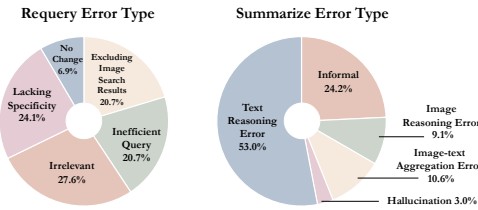

Figure 6: Distribution of Error types of GPT-4o (OpenAI, 2024b) and Qwen2-VL-7B (Qwen Team, 2024) in the end-to-end task.

Figure 7: Distribution of Error types of Qwen2-VL-7B (Qwen Team, 2024) in the reuqery and summarization task respectively.

## 3.3 ERROR ANALYSIS

To investigate the limitations of current LMM search capabilities, we conducted a comprehensive analysis of error types observed in our evaluation. Our proposed step-wise evaluation strategy enables analysis of failure modes for each core search step, complementing the end-to-end assessment. This analysis encompasses the entire benchmark. We first examine the end-to-end error types for both the best-performing closed-source model (GPT-4o) and open-source model (Qwen2-VL-7B). To better understand the failure cases, we then identify distinct error types in the requery and summarization task, which requires open-ended generation. We quantify these error types for a systematic understanding of current LMM limitations and point out critical areas for improvement.

**Error Analysis of End-to-end Task** In this paragraph, we are trying to answer the question: *Which step does LMM make a mistake in the end-to-end evaluation?* In Fig. 6, we showcase the statistics of different error types occurring in GPT-4o and Qwen2-VL-7B. We define the following four error categories: (i) *requery*, where the model requery is incorrect, and leads to all retrieved websites being invalid; (ii) *rerank*, where the model selects a website without a correct answer; (iii) *summarization*, where the full website content contains the information of correct answer, but the model fails to extract it; (iv) *informal*, the output format deviates from the prompt specifications. As shown in the figure, GPT-4o's primary error sources are rerank and summarization errors, while requery and informal errors account for approximately half the frequency of the main error causes. This suggests that GPT-4o's limitations lie primarily in information source ranking and multimodal information integration. As for Qwen2-VL, all four error types occur with similar frequency. The rise of the informal error portion may be attributed to the model's inferior instruction-following ability. Besides, it should be noted that the requery task demands advanced comprehension and key image information extraction ability. This task seldom appears in the training data of current LMMs. The prevalence of this error type in Qwen2-VL may indicate that it fails to generalize to adequately address this complex task.

**Error Analysis of Reuqery and Summarization Task** To better understand how open-source LMM makes the mistake, we dive into the requery and summarization task to find out the error patterns of Qwen2-VL-7B. We particularly select the two tasks requiring open-ended generation, which provides more information to identify the error. We define five types of error for both the requery and summarization tasks in Section E.2 and showcase the statistics in Fig. 7. The requery errors suggest that LMM often fails to fully understand the requery task and aggregate all available information. Besides, inefficient query error indicates that LMM has no clue about the real working scenario and query principles of search engines.

## 4 CONCLUSION

In this paper, we investigate the potential of LMMs as multimodal AI search engines. We first design MMSEARCH-ENGINE, a streamlined pipeline, enabling zero-shot LMMs to perform multimodal searches. To comprehensively assess the search capabilities, we introduce MMSEARCH, a benchmark comprising 300 queries across 14 subfields. Our evaluation methodology analyzes LMM search abilities step-by-step, facilitating a deeper understanding of their limitations. Using MMSEARCH-ENGINE, we evaluate various closed-source and open-source LMMs, revealing that current models still fall short of human-level search proficiency. Through thorough error analysis, we identify specific patterns of failure in key search process steps, providing valuable insights for future improvements in LMM search ability.

## ACKNOWLEDGEMENT

This project is funded in part by National Key R&D Program of China Project 2022ZD0161100, by the Centre for Perceptual and Interactive Intelligence (CPII) Ltd under the Innovation and Technology Commission (ITC)'s InnoHK, by NSFC-RGC Project N_CUHK498/24. Hongsheng Li is a PI of CPII under the InnoHK.

## REPRODUCIBILITY STATEMENT

We provide the demo code of MMSEARCH-ENGINE, which can be also used for inference, in the *code* directory in the supplementary material. We also include the end-to-end data in the *data* directory in the supplementary material, along with the loading script. Implementation details of inference settings are demonstrated in Section 3.1 and Section F.

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

SUPPLEMENTARY MATERIAL OVERVIEW

# A RELATED WORK

**Large Multimodal Models.** Recently, multimodal models (Radford et al., 2021; Li et al., 2022; OpenAI, 2023c; Rombach et al., 2022; Jiang et al., 2024c; Zong et al., 2024a) has gained unparalleled attention. Building on the success of Large Language Models (LLMs) (Touvron et al., 2023a;b) and large-scale vision models (Radford et al., 2021), Large Multimodal Models (LMMs) are gaining prominence across diverse domains. These models extend LLMs to handle tasks involving various modalities, including mainstream 2D image processing (Liu et al., 2023a; Zhu et al., 2023; Lin et al., 2023; Gao et al., 2023b; Ma et al., 2025), as well as 3D point clouds (Xu et al., 2023; Guo et al., 2023; 2024; Tang et al., 2025; Guo et al., 2025a), medical images (Xia et al., 2024c;b), videos (Li et al., 2023; Chen et al., 2023a; Zhang et al., 2023; Fu et al., 2024), reasoning Guo et al. (2025c); Jiang et al. (2025), and robotics Liu et al. (2024b); Jia et al. (2024). Among these LMMs, OpenAI's GPT-4o (OpenAI, 2024b) and Anthropic's Claude 3.5 Sonnet (Anthropic, 2024) demonstrate outstanding visual reasoning and comprehension capability, setting new standards in multi-modal performance. However, their closed-source nature limits broader adoption and development. In contrast, another research trajectory focuses on open-source LMMs for the community. Pioneering works like LLaVA (Liu et al., 2023a; 2024a; Li et al., 2024c;b), LLaMA-Adapter (Zhang et al., 2024b; Gao et al., 2023b), and MiniGPT-4 (Zhu et al., 2023; Chen et al., 2023b) incorporate a frozen CLIP (Radford et al., 2021) model for image encoding and integrate visual information into LLM for multi-modal instruction tuning. Later, works such as mPLUG-Owl (Ye et al., 2023a;b; 2024), SPHINX (Gao et al., 2024; Lin et al., 2023), and InternLM-XComposer (Dong et al., 2024) further advanced the field by incorporating diverse visual instruction tuning data and generalizing to more scenarios. More recent developments in the field have taken diverse directions. For example, several studies (Zong et al., 2024b; Tong et al., 2024) explore multiple vision encoders design. Meanwhile, other works (Liu et al., 2024a; Chen et al., 2024d; Qwen Team, 2024) incorporate high-resolution image input. Multi-image instruction data (Li et al., 2024c; Jiang et al., 2024b) is also integrated to enable perception across multiple images. While various benchmarks, both in the general (Fu et al., 2023; Jiang et al., 2025; Liu et al., 2023b; Yu et al., 2023; Xia et al., 2024a; Hong et al., 2025) and expert (Zhang et al., 2024c; Lu et al., 2023; 2022; Xia et al.) domain, has been proposed, the potential of LMM to function as a multimodal search engine remains largely unexplored. To this end, we introduce the MMSEARCH benchmark, which evaluates LMMs' zero-shot abilities of multimodal search, offering valuable insights for future research.

**Large models with Retrieval Augmented Generation (RAG).** RAG (Retrieval-Augmented Generation) is an effective strategy for enhancing model knowledge by retrieving relevant information from external sources (Fan et al., 2024). RAG has been leveraged in various scenarios including knowledge-intensive question answering (Borgeaud et al., 2022; Guu et al., 2020), machine translation (He et al., 2021), and hallucination elimination (Béchard & Ayala, 2024). Current works has focused on improving specific aspects of RAG. RG-RAG (Chan et al., 2024) proposes to refine the query for retrieval by decomposition and disambiguation. Self-RAG (Asai et al., 2023) incorporates the self-reflection of LLM to enhance the generation quality. The AI search engine could be viewed as a form of RAG with the Internet serving as the external knowledge source. Recently, MindSearch (Chen et al., 2024c) proposes an AI search engine framework to simulate the human minds in web information seeking. Meanwhile, multiple benchmarks of RAG (Yang et al., 2024b;

Chen et al., 2024b) have been introduced to comprehensively evaluate a RAG system. However, both the current AI search engine and RAG benchmark are limited to the text-only setting, leaving the multimodal search engine and evaluation largely unexplored. To bridge this gap, we introduce MMSEARCH-ENGINE and MMSEARCH, a multimodal AI search engine pipeline and dataset designed to evaluate various multimodal scenarios.

# B  AUTOMATED DATA CURATION PIPELINE

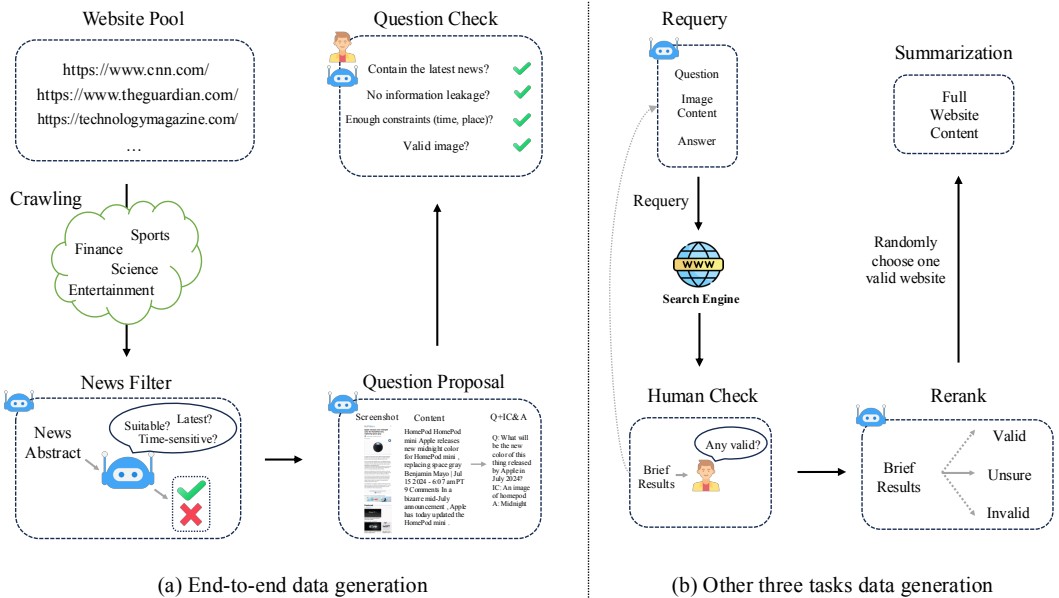

(a) End-to-end data generation      (b) Other three tasks data generation

Figure 8: **Illustration of the Data Curation Pipeline.** We first collect the end-to-end data by crawling inside the website pool and prompting LMMs to raise questions based on the content. Then we further prompt the LMM to generate annotation for other three tasks. The human check is optional for the end-to-end data generation but compulsory for other three tasks data generation.

Now we introduce our automated/semi-automated data curation pipeline. The figure is shown in Fig. 8. We first define a website pool and a model pool. The website pool contains general news websites like CNN and expertise websites like Arxiv. The model pool contains state-of-art models for the data curation pipeline to guarantee diversity and fairness.

1. **End-to-end data curation.** We employ a web crawler to obtain all the sub-websites published later than a specific date. However, not all the websites are suitable for raising a question to test the LMMs' searching capability. For example, some websites do not contain any recent news, while some websites' contents are difficult to convert into a question with a definite answer. Therefore, we randomly choose a model from the model pool to serve as a news filter, prompting it to filter valid websites by providing some few-shot examples. Next, we provide the text contents and screenshots of the valid websites to a model from the model pool. The model is asked to raise several questions based on the website content. It is encouraged to raise questions unable to be answered only by text. As for the question with an image, the model is asked to briefly describe the image content, and we will later use the description to search in Bing and obtain the first result image. Finally, we apply the quality check of the generated questions and their corresponding images either by human or a model from the model pool.

2. **Rerank data generation.** We provide the generated requery to the search engine and retrieve K websites for rerank. Again, we provide the question, the question image content, and the answer to a model and ask to categorize each website into valid, unsure or invalid.

3. **Summarization data generation.** We randomly choose one website from the websites marked as valid in the last step and obtain its full content.

Table 4: **Evaluation Results of Different Complexity in MMSᴇᴀʀᴄʜ.** We categorize all the questions into three levels of complexity.

| Model | All | | | End-to-end | | | Requery | | | Rerank | | | Summarization | | |
|---|---|---|---|---|---|---|---|---|---|---|---|---|---|---|---|
| | Easy | Middle | Hard | Easy | Middle | Hard | Easy | Middle | Hard | Easy | Middle | Hard | Easy | Middle | Hard |
| *Closed-source LMMs with MMSᴇᴀʀᴄʜ-Eɴɢɪɴᴇ* | | | | | | | | | | | | | | | |
| Claude 3.5 Sonnet (Anthropic, 2024) | 56.2 | 53.6 | 48.9 | 53.0 | 49.8 | 45.0 | 49.3 | 37.6 | 35.0 | 80.3 | 84.5 | 75.3 | 59.8 | 59.9 | 58.3 |
| GPT-4V (OpenAI, 2023c) | 55.3 | 56.4 | 53.1 | 52.7 | 52.9 | 50.2 | 52.7 | 41.4 | 38.8 | 76.5 | 82.8 | 80.2 | 55.0 | 63.9 | 54.3 |
| GPT-4o (OpenAI, 2024b) | 63.1 | 63.9 | 59.2 | 61.0 | 62.3 | 57.4 | 54.3 | 41.2 | 40.6 | 82.2 | 85.6 | 81.5 | 64.0 | 65.5 | 59.0 |
| *Open-source LMMs with MMSᴇᴀʀᴄʜ-Eɴɢɪɴᴇ* | | | | | | | | | | | | | | | |
| Mantis (Jiang et al., 2024b) | 22.8 | 18.0 | 13.0 | 19.4 | 15.9 | 10.0 | 28.5 | 12.6 | 14.5 | 40.5 | 43.1 | 34.6 | 28.2 | 11.5 | 12.9 |
| InternLM-XC2.5 (Zhang et al., 2024a) | 26.6 | 19.3 | 18.1 | 28.5 | 20.0 | 16.7 | 24.3 | 26.0 | 25.0 | 0.0 | 0.0 | 0.0 | 39.5 | 30.3 | 42.7 |
| InternLM-XC2.5$_{AnyRes}$ | 25.6 | 18.1 | 21.2 | 27.5 | 18.3 | 21.4 | 20.2 | 24.6 | 20.9 | 0.0 | 0.0 | 0.0 | 39.5 | 31.5 | 41.4 |
| LLaVA-NeXT-Interleave (Li et al., 2024c) | 33.2 | 25.6 | 23.3 | 28.5 | 20.2 | 16.9 | 31.0 | 22.1 | 22.7 | 61.4 | 51.7 | 49.4 | 41.5 | 41.6 | 45.0 |
| mPlug-Owl3 (Ye et al., 2024) | 37.1 | 28.5 | 27.7 | 30.5 | 20.6 | 19.5 | 40.8 | 28.3 | 23.9 | 77.7 | 73.0 | 70.4 | 44.8 | 44.1 | 48.4 |
| mPlug-Owl3$_{AnyRes}$ | 37.5 | 32.8 | 29.0 | 31.4 | 26.4 | 21.5 | 38.6 | 26.9 | 26.0 | 76.5 | 77.6 | 67.9 | 44.0 | 39.6 | 48.3 |
| InternVL2 (Chen et al., 2024d) | 38.7 | 30.9 | 30.5 | 35.0 | 28.3 | 26.8 | 39.5 | 25.3 | 27.9 | 57.6 | 38.5 | 37.0 | 47.4 | 46.2 | 52.8 |
| InternVL2$_{AnyRes}$ | 38.0 | 30.0 | 32.0 | 34.6 | 24.5 | 28.2 | 38.8 | 24.5 | 26.5 | 53.8 | 59.8 | 45.1 | 46.7 | 43.9 | 50.2 |
| Idefics3 (Laurençon et al., 2024) | 43.1 | 30.5 | 31.0 | 38.0 | 21.3 | 23.5 | 40.9 | 23.3 | 23.3 | 76.5 | 83.3 | 69.1 | 48.9 | 50.0 | 53.1 |
| Idefics3$_{AnyRes}$ | 43.2 | 31.2 | 28.4 | 38.4 | 24.1 | 23.0 | 37.6 | 20.3 | 17.6 | 76.5 | 74.7 | 64.2 | 48.4 | 46.6 | 38.4 |
| LLaVA-OneVision (Li et al., 2024b) | 42.9 | 32.3 | 31.1 | 36.8 | 24.6 | 23.3 | 46.5 | 28.9 | 25.8 | 78.0 | 67.8 | 69.8 | 51.5 | 56.7 | 53.6 |
| Qwen2-VL$_{AnyRes}$ (Qwen Team, 2024) | 45.6 | 44.9 | 45.3 | 41.2 | 38.9 | 40.4 | 46.5 | 34.0 | 32.2 | 73.5 | 83.3 | 74.7 | 50.3 | 56.8 | 59.6 |
| LLaVA-OneVision (72B) | 52.2 | 48.0 | 49.0 | 47.6 | 41.6 | 44.0 | 51.0 | 39.6 | 33.2 | 79.5 | 85.1 | 83.3 | 60.3 | 63.7 | 60.6 |
| Qwen2-VL$_{AnyRes}$ (72B) | 53.7 | 53.4 | 50.2 | 50.7 | 48.6 | 46.9 | 52.3 | 40.0 | 37.1 | 72.3 | 82.2 | 77.8 | 58.5 | 67.0 | 53.6 |

Notably, human check is a must for the requery data generation process. There should be at least one valid website to guarantee the effectiveness of the generated requery. Only after human check of this step, the quality of rerank and summarization data generation is assured.

## C  ADDITIONAL DATASET DETAILS

We manually annotate the complexity of the data based on the difficulty of the three steps in MMSearch-Engine:

1. **Requery difficulty.** This concerns the complexity of transforming the original question into an effective search query. Complex cases arise when the question references image content, requiring the LMM to first analyze the visual information and then synthesize it with the text question into a coherent search query. For instance, when a user asks about a landmark shown in an image, the LMM must first identify the landmark through image search and then incorporate this information into a text query about the landmark's specific attributes.

2. **Rerank difficulty.** This dimension evaluates the challenge of identifying and prioritizing relevant search results. The difficulty primarily scales with the information scarcity. If there are only very limited websites containing useful information, it is more difficult to successfully retrieve and choose the website.

3. **Summarization difficulty.** This aspect involves both information synthesis and multi-modal reasoning challenges. In cases requiring synthesis, answers cannot be derived from a single source sentence - the LMM must integrate information scattered across different parts of the website. For example, comparing event frequencies across locations (like concert counts between cities) requires gathering and analyzing distributed data. Additionally, some questions demand analysis of both textual and visual website content, sometimes necessitating comparison with input images with images in the website.

Based on these criteria, we have categorized all questions into three difficulty levels, with the following distribution: hard (28%), medium (27.7%), and easy (44.3%). We provide the evaluation results grouped by the difficulty levels in Table 4.

## D FUTURE DIRECTION

Our proposed MMSEARCH-ENGINE can be enhanced through interactive user feedback loops. When the model produces an incorrect answer, users can identify the specific step where the error occurred and prompt the model to reconsider its reasoning. This iterative process allows for guided model refinement until accurate results are achieved. We conducted preliminary experiments on GPT-4o with this approach on three test cases. The results demonstrated that the LMM successfully interpreted user feedback and appropriately adjusted its responses based on the provided guidance. The model's ability to understand the user input and revise its reasoning suggests significant potential for improving task accuracy.

## E ADDITIONAL EXPERIMENTS AND ANALYSIS

### E.1 SCALING TEST-TIME COMPUTE VS SCALING MODEL SIZE

Recent works such as OpenAI o1 (OpenAI, 2024a) and Li et al. (2024d) have highlighted the critical role of scaling test-time computation in enhancing model performance. Our end-to-end task, which requires multiple Internet interactions, presents an opportunity to investigate the potential of scaling test-time computation compared to scaling model size. To explore this, we conduct experiments using LLaVA-OneVision-7B (Li et al., 2024b), focusing on scaling test-time computation, and compare against LLaVA-OneVision-72B scaling in model size, which aims to provide insights into the relative benefits of increased inference computation versus increased model parameters.

For scaling up the test-time computation, we adopt a multi-modal search strategy similar to best-of-N solution, where 'N' denotes 25 in our settings. Specifically, for LLaVA-OneVision-7B, we first prompt the model to generate a requery 5 times, from which we selected the one with the highest requery score $S_{req}$. This requery is then used to retrieve brief results from 8 websites from a search engine. The model is again prompted 5 times to select the most informative website. After removing duplicates from the selected websites, we extract the full website content from the remaining ones and prompt the

Table 5: **Scaling Test-Time Compute vs Scaling Model Size.** 'TTC' and $S_{e2e}$ denote Test-Time Computation and the score of end-to-end task.

| Model | Inference Cost | $S_{e2e}$ |
|---|---|---|
| LLaVA-OV-7B | 1 | 29.6 |
| LLaVA-OV-7B (TTC) | $\sim$25 | 55.2 |
| LLaVA-OV-72B | $\sim$6 | 44.9 |

model to answer 5 times, obtaining 25 end-to-end outputs in total. We compute the F1 score for each answer against the ground truth and take the maximum as the model's end-to-end score for the query. Table 5 shows that LLaVA-OneVision-7B (TTC) achieves the score of 55.2% in the end-to-end task, significantly enhancing the original score of 29.6%, which surpasses LLaVA-OneVision-72B's 44.9% and GPT-4V's 52.1%. This result reveals the substantial potential of scaling test-time computation, validating the effectiveness of this technique as introduced by OpenAI o1. Our findings provide valuable insights for future research in this domain, suggesting that increased inference computation may offer comparable or superior performance improvements to increased model size not only in math and code tasks, but also in multimodal search tasks.

### E.2 DEFINITION OF ERRORS IN THE REQUERY AND SUMMARIZATION TASKS

Five types of requery error:

- *Lacking Specifility*, where the model fails to include all the specific information in the requery and therefore leads to sub-optimal search results. For example, the query is asking the release date of Vision Pro in China. However, the model omits the condition of China and directly asks about the release date of Vision Pro.

- *Inefficient Query*, where the model does not consider the real scenario and the requery is inefficient for the search engine to find the answer. For example, the query is asking whether the Van Gogh's Sunflowers and Antoni Clavé's Grand Collage are both oil paintings. Clearly, it is a commonsense that Van Gogh's Sunflowers is an oil painting and Antoni Clavé's Grand Collage is much less well-known. An efficient query should be asking about the images of Antoni Clavé's Grand Collage and further determine if it is also an oil painting by directly looking at it. However, the model directly asks the original query to the

search engine. There is very little chance that an exact same question has ever been raised so probably this requery will bring very little helpful information.

- *Excluding Image Search Results*, where the model totally ignores the information in the screenshot of the image search results and therefore lacks important specific information in the requery. For example, the query is 'When did this football player obtain the gold medal?' and provides an image of the player. The model is supposed to find out the player's name by viewing the image search result and raise a requery like '[PLAYER NAME] obtained the gold medal time'. However, the model fails to incorporate the player's name in the requery and definitely the retrieved websites will not include any helpful information.

- *No Change*, where the model just uses the question as the query input to the search engine.

- *Irrelevant*, where the model either matches wrong information from the image search result or mistakenly understands the query and outputs an irrelevant requery.

Five types of summarization error:

- *Text Reasoning Error*, where the model fails to extract the answer from the website textual information.

- *Image-text Aggregation Error*, where obtaining the answer needs combining the information from both images and texts. The model fails to do so.

- *Image reasoning Error*, where the model fails to extract the answer from the image, and the answer can only be obtained from the image.

- *Hallucination* (Huang et al., 2023), where the model provides an unfaithful answer that cannot be grounded in the given content.

- *Informal*, the output format does not follow the prompt specifications, the same error type in the end-to-end task.

## F ADDITIONAL EXPERIMENTAL DETAILS

**Rationality of the weight in final score.** Considering the sequential nature of the tasks, the reason for our weighting scheme includes two complementary perspectives:

1. The importance of each task due to the cascaded nature is already reflected in the end-to-end score. Detailedly, although it is easy to discern that the upstreamed task is more important, it is difficult to assign a precise weight to each of them. So we do not manually assign the weights but directly focus mainly on the end-to-end score, which implicitly considers their cascaded nature.

2. The individual task weights serve as complementary metrics rather than indicators of relative importance. Relying solely on end-to-end evaluation, while comprehensive, may obscure the performance characteristics of individual components and hinder targeted improvements. We therefore maintain independent evaluation of each task, with the weight distribution designed to balance the prominence of the end-to-end metric against the component-level assessments. This dual evaluation strategy enables both system-level optimization and component-specific refinements.

**More Implementation Details** All our experiments of open-source models are conducted without any fine-tuning on search data or tasks. As for the prompts, the requery prompt contains 3 examples to better guide LMMs to output a valid requery. While prompts for other tasks are all in a zero-shot setting. We prompt the LMM to output as few words as possible for a better match with the ground truth. We employ the metric introduced in Section 2.3. Besides, we recruit eight qualified college students and ask them to solve the problems in MMSEARCH independently, following the same pipeline of MMSEARCH-ENGINE. This score serves as a baseline for human performance. We conduct all experiments on NVIDIA A100 GPUs.

The input image dimensions for the webpage's top section screenshot were set to $1024 \times 1024$ pixels. For the full-page screenshot, we set the initial webpage width to $512$ pixels, although the actual width of a small portion of webpages may vary due to its layout settings. Furthermore, considering that a

full-page screenshot can be extremely lengthy, directly inputting it as a single image into an LLM would result in excessive downsizing, making the content too vague for accurate identification. To address this, we segmented the full-page screenshot into multiple images, starting from the top, with each segment measuring 512 pixels in height. Because of the context length limitations of LMMs, the maximum number of full-page screenshot segments is therefore restricted to 10.

**Full-page Screenshot Slimming.** For the full-page screenshot, we compute the Sobel gradients (Kanopoulos et al., 1988) to detect the edges and generate a gradient magnitude image. We iteratively remove the areas with gradients below a threshold, which represent the blank areas. This approach, shown in Fig. 9, effectively reduces image size while maintaining the document content.

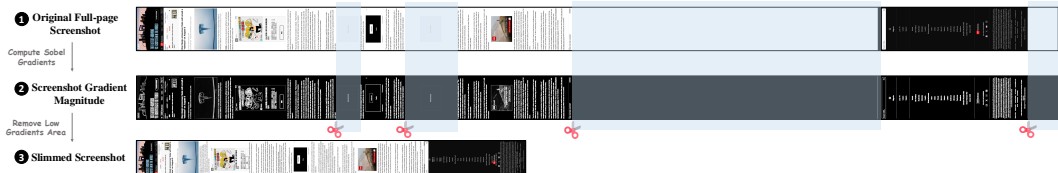

Figure 9: **Illustration of the Screenshot Slim Process.** We leverage Sobel gradients (Kanopoulos et al., 1988) to identify blank areas and remove them. After slimming, the screenshot size is largely reduced without any information loss.

**Model Sources.** For different LMMs, we select their latest models with size around 7B for evaluation to fully reveal their multimodal search proficiency. Table 6 presents the release time and model sources of LMMs used in MMSEARCH.

Table 6: **The Release Time and Model Source of LMMs Used in MMSEARCH.**

| Model | Release Time | Source |
|---|---|---|
| GPT-4V (OpenAI, 2023c) | 2023-09 | https://platform.openai.com/ |
| GPT-4o (OpenAI, 2024b) | 2024-05 | https://platform.openai.com/ |
| Claude 3.5 Sonnet (Anthropic, 2024) | 2024-06 | https://www.anthropic.com/news/claude-3-5-sonnet |
| InternLM-XC2.5 (Zhang et al., 2024a) | 2024-07 | https://github.com/InternLM/InternLM-XComposer |
| Mantis (Jiang et al., 2024b) | 2024-05 | https://tiger-ai-lab.github.io/Mantis/ |
| LLaVA-NeXT-Interleave (Li et al., 2024c) | 2024-06 | https://github.com/LLaVA-VL/LLaVA-NeXT |
| InternVL2 (Chen et al., 2024d) | 2024-07 | https://github.com/OpenGVLab/InternVL |
| mPlug-Owl3 (Ye et al., 2024) | 2024-08 | https://github.com/X-PLUG/mPLUG-Owl |
| Idefics3 (Laurençon et al., 2024) | 2024-08 | https://huggingface.co/HuggingFaceM4/Idefics3-8B-Llama3 |
| LLaVA-OneVision (Li et al., 2024b) | 2024-08 | https://llava-vl.github.io/blog/2024-08-05-llava-onevision/ |
| Qwen2-VL (Qwen Team, 2024) | 2024-08 | https://github.com/QwenLM/Qwen2-VL |

**Input Prompts of LMM for Response Generation.** We showcase the input prompts of LMM for the three tasks respectively in Table 7-9. We adopt two types of prompts for queries with an image and without images. For query with an image, we specifically require the LMM to leverage the image search result to solve the task.

Table 7: **Input Prompt of LMMs for Requery.** We adopt two different prompts for the query with image input and without image input. We leverage a 3-shot prompt to guide the LMM to generate a reasonable requery.

| Question | Prompt |
| --- | --- |
| Query without image | You are a helpful assistant. I am giving you a question, which cannot be solved without external knowledge. Assume you have access to a text-only search engine (e.g., google). Please raise a query to the search engine to search for what is useful for you to answer the question correctly. Your query needs to consider the attribute of the query to search engine. Here are 3 examples:
Question: Did Zheng Xiuwen wear a knee pad in the women's singles tennis final in 2024 Paris Olympics? Query to the search engine: Images of Zheng Xiuwen in the women's singles tennis final in 2024 Paris Olympics
Question: When will Apple release iPhone16? Query to the search engine: iPhone 16 release date
Question: Who will sing a French song at the Olympic Games closing ceremony? Query to the search engine: Singers at the Olympic Games closing ceremony, French song.
Question: $\{question\}$.
Query to the search engine (do not involve any explanation): |
| Query with image | You are a helpful assistant. I am giving you a question including an image, which cannot be solved without external knowledge. Assume you have access to a search engine (e.g., google). Please raise a query to the search engine to search for what is useful for you to answer the question correctly. You need to consider the characteristics of asking questions to search engines when formulating your questions. You are also provided with the search result of the image in the question. You should leverage the image search result to raise the text query. Here are 3 examples:
Question: Did Zheng Xiuwen wear a knee pad in the women's singles tennis final in 2024 Paris Olympics? Query to the search engine: Images of Zheng Xiuwen in the women's singles tennis final in 2024 Paris Olympics
Question: When will Apple release iPhone16? Query to the search engine: iPhone 16 release date
Question: Who will sing a French song at the Olympic Games closing ceremony? Query to the search engine: Singers at the Olympic Games closing ceremony, French song
Question: $\{query\_image\}\{question\}$. The image search result is: $\{image\_search\_result\}$
Query to the search engine (do not involve any explanation): |

Table 8: **Input Prompt of LMMs for Rerank.** We adopt two different prompts for the query with image input and without image input.

| Question | Prompt |
|---|---|
| Query without image | You are a helpful assistant. I am giving you a question and 8 website information related to the question (including the screenshot, snippet and title). You should now read the screenshots, snippets and titles. Select 1 website that is the most helpful for you to answer the question. Once you select it, the detailed content of them will be provided to help you correctly answer the question. The question is $\{question\}$. The website informations is $\{website\_information\}$. 

 You should directly output 1 website's index that can help you most, and enclose the website in angle brackets. The output format should be: $<$Website Index $>$. An example of the output is: $<$Website 1 $>$. Your answer: |
| Query with image | You are a helpful assistant. I am giving you a question including an image. You are provided with the search result of the image in the question. And you are provided with 8 website information related to the question (including the screenshot, snippet, and title). You should now read the screenshots, snippets and titles of these websites. Select 1 website that is the most helpful for you to answer the question. Once you select it, the detailed content of them will be provided to help you correctly answer the question. The question is $\{query\_image\}\{question\}$. The image search result is $\{image\_search\_result\}$. The website information is $\{website\_information\}$. 

 You should directly output 1 website's index that can help you most, and enclose the website in angle brackets. The output format should be: $<$Website Index $>$. An example of the output is: $<$Website 1 $>$. Your answer: |

Table 9: **Input Prompt of LMMs for Summarization.** We adopt two different prompts for the query with image input and without image input.

| Question | Prompt |
| --- | --- |
| Query without image | You are a helpful assistant. I am giving you a question and 1 website information related to the question. Please follow these guidelines when formulating your answer: 1. If the question contains a false premise or assumption, answer "invalid question". 2. When answering questions about dates, use the yyyy-mm-dd format. 3. Answer the question with as few words as you can.

You should now read the information of the website and answer the question. The website information is $\{website\_information\}$. The question is $\{question\}$. Please directly output the answer without any explanation: |
| Query with image | You are a helpful assistant. I am giving you a question including an image. You are provided with the search result of the image in the question. And you are provided with 1 website information related to the question. Please follow these guidelines when formulating your answer: 1. If the question contains a false premise or assumption, answer "invalid question". 2. When answering questions about dates, use the yyyy-mm-dd format. 3. Answer the question with as few words as you can.

You should now read the information of the website and answer the question. The website information is $\{website\_information\}$. The image search result is $\{image\_search\_result\}$. The question is $\{query\_image\}\{question\}$. Please directly output the answer without any explanation: |

# G  MORE DATA DETAILS

## G.1  DATA EXAMPLE OF 4 EVALUATION TASKS

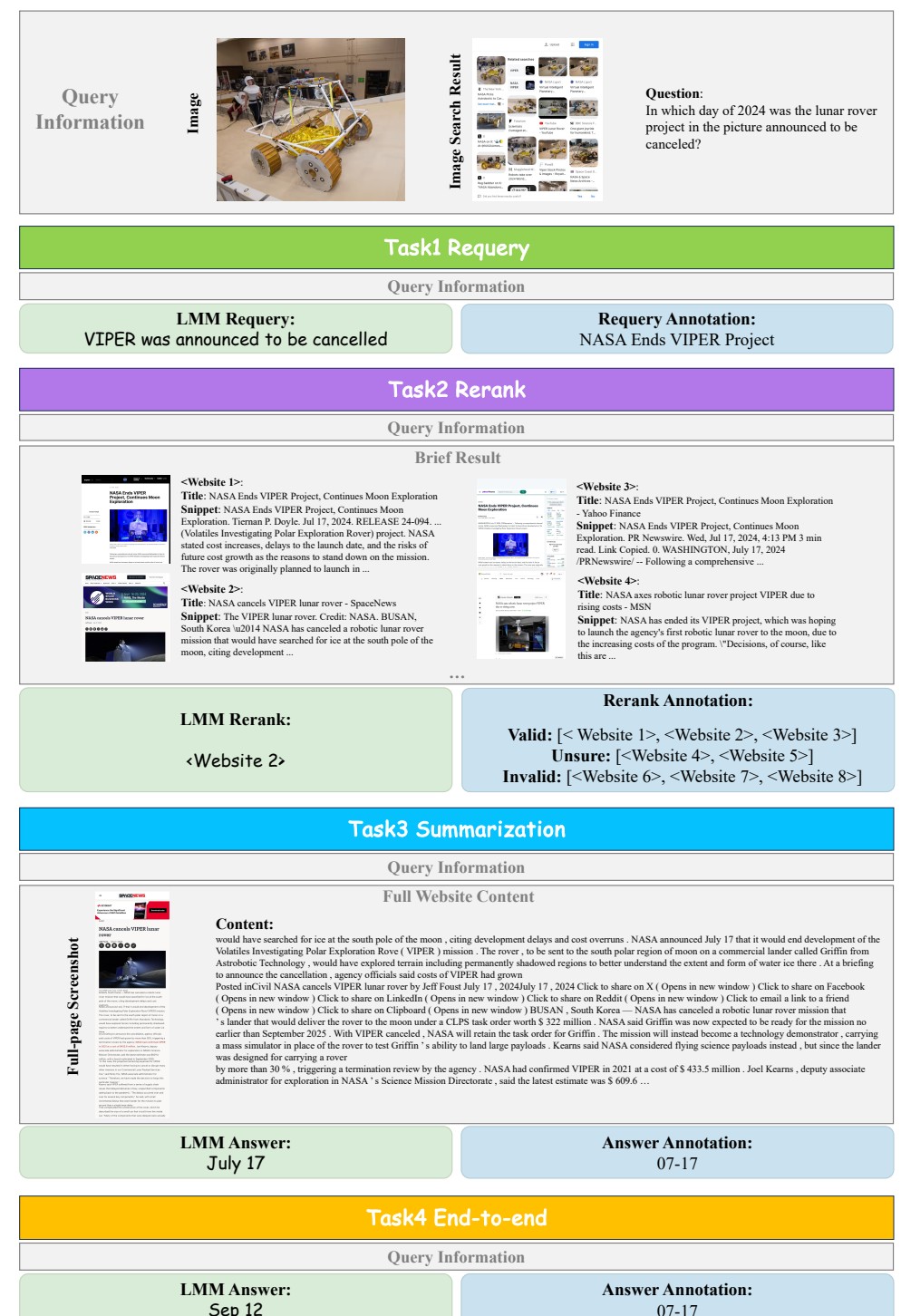

Figure 10: **Example Input, LMM Output, and Ground Truth for Four Evaluation Tasks.** The color-coding of each module corresponds to Fig. 4. Task1 Requery (green), Task2 Rerank (purple), Task3 Summarization (blue), and Task4 End-to-end (yellow) are shown. Image best viewed in color.

## G.2  Subfield definition

**News** area encompasses a vast spectrum of information, ranging from everyday events to engaging entertainment content and specialized fields such as scientific discoveries and financial analysis. This comprehensive coverage serves as a rigorous assessment of the model's ability to process information in diverse domains. We divide this expansive area into eight distinct subfields:

- **Traditional Sports**: Data concerning traditional athletic competitions, team performances, player statistics, and sporting events. This includes scores, league standings, player transfers, and analysis of various professional sports across different leagues and countries.

- **e-Sports**: Information about competitive video gaming, including tournament results, player rankings, and league information. This covers various game titles, team formations, streaming viewership statistics, and tournament information.

- **Technology**: Information about technological innovations, gadgets, software developments, and tech industry news. This includes product launches, software updates, cybersecurity issues, and artificial intelligence advancements.

- **Paper**: Content related to academic papers, research publications, and scholarly articles in various artificial intelligence fields. The queries include method explanation, figure understanding, and experiment settings.

- **Entertainment**: Data about movies, music, television, celebrities, and other forms of popular entertainment. It also includes data concerning video games.

- **Finance**: Information on financial markets, economic indicators, business news, and monetary policies. This covers stock prices, company earnings reports, company financial statements, and regulatory news regarding finance.

- **General News**: Broad coverage of various news topics not specific to any particular subfield. This includes a mix of local and global events, human interest stories, lifestyle articles, climate news, and general interest content that doesn't fit neatly into other specialized news subfields.

- **False Premise**: Data related to misinformation or incorrect assumptions in the query. This subfield focuses on fact-checking capabilities. All the answers to the queries of this subfield are 'invalid question'.

**Knowledge** area represents broad subfields of information and data related to general knowledge across various disciplines. This area concentrates on rare knowledge that most LMMs fail to answer. We categorize this area into five subfields:

- **Architecture**: Information about building design, architectural styles, building information, and construction projects. This includes city landmarks, the comparison of architectural styles, and multi-view architecture matchings.

- **Arts**: Data concerning visual arts, drawings, sculptures, badges, and other forms of creative expression. This covers artwork details, artist profiles, artwork history, and artwork style comparisons.

- **Fashion**: Content related to clothing trends, fashion brands, and designer collections. This includes retail price, clothing style, release date, and brand information.

- **Astronomy**: Information about celestial objects, space exploration, astronomical phenomena, and related research. This covers observational data from telescopes and image results from space missions. The questions focus on the background information of these celestial objects presented in the query image.

- **Anime**: Data about Japanese animation, including series storylines and character information. This encompasses character background, character appearance, voice actor information, and chapter information.

- **Auto**: Content related to automobiles, including vehicle specifications, industry trends, and automotive technology. This covers new car models, performance test results, coefficients of cars, and release date.

## H   QUALITATIVE EXAMPLES

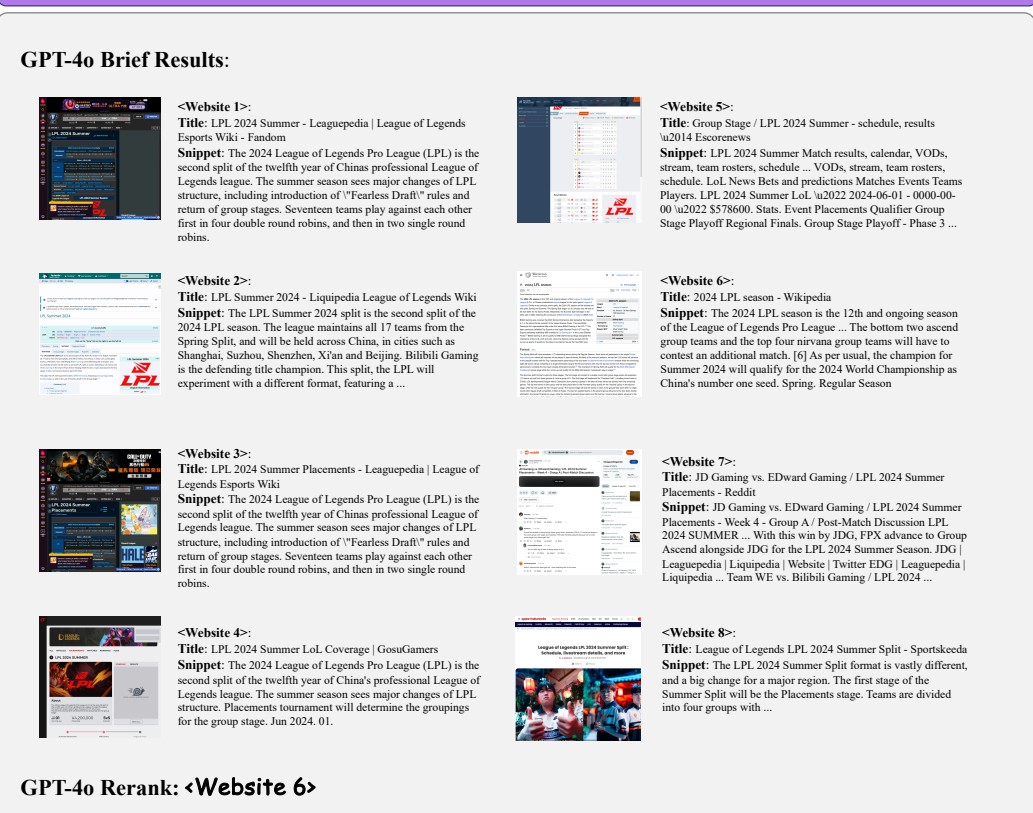

Figure 11: Response and middle results comparison of GPT-4o (OpenAI, 2024b), Qwen2-VL-7B (Qwen Team, 2024), and LLaVA-OneVision-7B (Li et al., 2024b) in the end-to-end task.

**Round2 Rerank**

**Qwen2-VL Brief Results**:

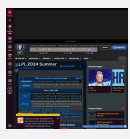

**<Website 1>**:
**Title**: LPL 2024 Summer - Leaguepedia | League of Legends Esports Wiki - Fandom
**Snippet**: The 2024 League of Legends Pro League (LPL) is the second split of the twelfth year of China's professional League of Legends league. The summer season sees major changes of LPL structure, including introduction of \"Fearless Draft\" rules and return of group stages. Seventeen teams play against each other first in four double round robins, and ...

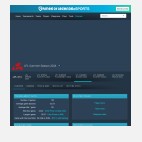

**<Website 5>**:
**Title**: LPL Summer Season 2024 stats - Games of Legends
**Snippet**: View all the stats for LPL Summer Season 2024: matches result, team ranking, best players, most played champions.

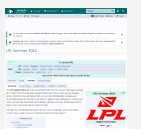

**<Website 2>**:
**Title**: LPL Summer 2024 - Liquipedia League of Legends Wiki
**Snippet**: The LPL Summer 2024 is the second split of the League of Legends Pro League season. Stay up to date with match results, schedules, and broadcasts here!

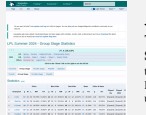

**<Website 6>**:
**Title**: LPL Summer 2024 - Group Stage Statistics - Liquipedia
**Snippet**: Liquipedia app major update: Revamped player and team pages with schedule, results, stats, achievements and more. Download the latest version on iOS or Android and read our update blog here.

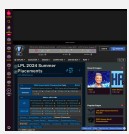

**<Website 3>**:
**Title**: LPL/2024 Season/Summer Placements - League of Legends Esports Wiki
**Snippet**: The 2024 League of Legends Pro League (LPL) is the second split of the twelfth year of Chinas professional League of Legends league. The summer season sees major changes of LPL structure, including introduction of \"Fearless Draft\" rules and return of group stages. Seventeen teams play against each other first in four double round robins, and then in two single round robins. The top ten teams ...

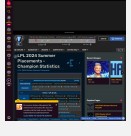

**<Website 7>**:
**Title**: LPL/2024 Season/Summer Placements/Champion Statistics
**Snippet**: Tournament: LPL/2024 Season/Summer Placements - Showing Values Per Game - Open As Query. Champion Statistics - 130 Total Games Played with 92 Champions Contested. Restrict By Role (Query Permalinks): Top.

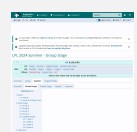

**<Website 4>**:
**Title**: LPL 2024 Summer - Group Stage - Liquipedia League of Legends Wiki
**Snippet**: The Group Stage Draw Show will be held on May 21, 2024 20:00 CST. [1] Teams are splited into 4 seed pools to ensure quality groups. Pool 4 will consist of the bottom 4 teams, namely: RNG, RA, EDG, UP. One team from Pool 1, 2 and 4 will be seeded to each group. One group will comprise of two Pool 3 teams, while the 3 other groups will have one ...

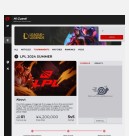

**<Website 8>**:
**Title**: LPL 2024 Summer LoL Coverage | GosuGamers
**Snippet**: The 2024 League of Legends Pro League (LPL) is the second split of the twelfth year of China's professional League of Legends league. The summer season sees major changes of LPL structure. Placements tournament will determine the groupings for the group stage. Jun 2024. 01.

**Qwen2-VL Rerank**: `<Website 1>`

**LLaVA-OneVision Brief Results**:

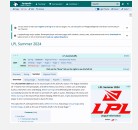

**<Website 1>**:
**Title**: LPL Summer 2024 - Liquipedia League of Legends Wiki
**Snippet**: The LPL Summer 2024 split is the second split of the 2024 LPL season. The league maintains all 17 teams from the Spring Split, and will be held across China, in cities such as Shanghai, Suzhou, Shenzhen, Xi'an and Beijing. Bilibili Gaming is the defending title champion. This split, the LPL will experiment with a different format, featuring a ...

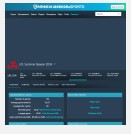

**<Website 5>**:
**Title**: LPL Summer Season 2024 stats - Games of Legends
**Snippet**: View all the stats for LPL Summer Season 2024: matches result, team ranking, best players, most played champions. LPL Summer Season 2024 stats: Team ranking, Top KDA, picks and bans. Sign In (Log In) ... LPL Summer Season 2024 stats; Tournament data; Number of games: 152: Average game duration: 32:57: Average kills / game: 28: Shortest game:

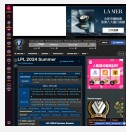

**<Website 2>**:
**Title**: LPL 2024 Summer - Leaguepedia | League of Legends Esports Wiki
**Snippet**: The 2024 League of Legends Pro League (LPL) is the second split of the twelfth year of Chinas professional League of Legends league. The summer season sees major changes of LPL structure, including introduction of \"Fearless Draft\" rules and return of group stages. Seventeen teams play against each other first in four double round robins, and then in two single round robins.

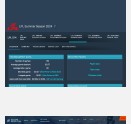

**<Website 6>**:
**Title**: Group Stage / LPL 2024 Summer - schedule, results - Escorenews
**Snippet**: LPL 2024 Summer Match results, calendar, VODs, stream, team rosters, schedule ... VODs, stream, team rosters, schedule. LoL News Bets and predictions Matches Events Teams Players. LPL 2024 Summer LoL \u2022 2024-06-01 - 0000-00-00 \u2022 $578600. Stats. Event Placements Qualifier Group Stage Playoff Regional Finals. Group Stage Playoff - Phase 3 ...

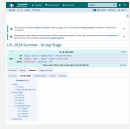

**<Website 3>**:
**Title**: LPL 2024 Summer - Group Stage - Liquipedia League of Legends Wiki
**Snippet**: The Group Stage Draw Show will be held on May 21, 2024 20:00 CST. [1] Teams are splited into 4 seed pools to ensure quality groups. Pool 4 will consist of the bottom 4 teams, namely: RNG, RA, EDG, UP. One team from Pool 1, 2 and 4 will be seeded to each group. One group will comprise of two Pool 3 teams, while the 3 other groups will have one ...

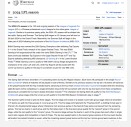

**<Website 7>**:
**Title**: 2024 LPL season - Wikipedia
**Snippet**: The 2024 LPL season is the 12th and ongoing season of the League of Legends Pro League ... The bottom two ascend group teams and the top four nirvana group teams will have to contest an additional match. [6] As per usual, the champion for Summer 2024 will qualify for the 2024 World Championship as China's number one seed. Spring. Regular Season

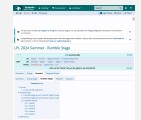

**<Website 4>**:
**Title**: LPL 2024 Summer - Rumble Stage - Liquipedia League of Legends Wiki
**Snippet**: Format. Rumble Stage. Single round robin within each group. All matches are Bo3 using Standard Draft. For the Group \"Ascend\" (High Group): Top 2 teams get seeded into the semifinals. 3rd and 4th place teams get seeded into the fourth round of playoffs. 5th and 6th place teams get seeded into the third round of playoffs.

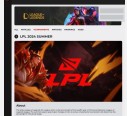

**<Website 8>**:
**Title**: LPL 2024 Summer LoL Coverage | GosuGamers
**Snippet**: The 2024 League of Legends Pro League (LPL) is the second split of the twelfth year of China's professional League of Legends league. The summer season sees major changes of LPL structure. Placements tournament will determine the groupings for the group stage. Jun 2024. 01.

**LLaVA-OneVision Rerank**: `<Website 4>`

Figure 12: Response and middle results comparison of GPT-4o (OpenAI, 2024b), Qwen2-VL-7B (Qwen Team, 2024), and LLaVA-OneVision-7B (Li et al., 2024b) in the end-to-end task.

## Round3 Summarization

**GPT-4o**

**Full-page Screenshot:**

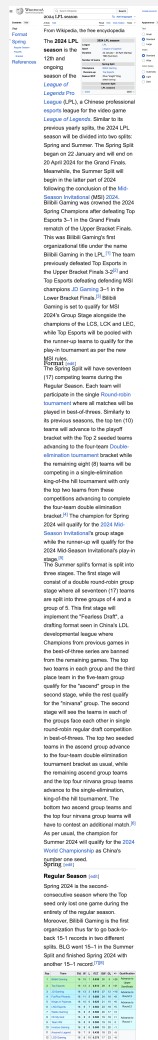

**Content:**

for Spring 2024 will qualify for the 2024 Mid-Season Invitational 's group stage while the runner-up will qualify for the 2024 Mid-Season Invitational 's play-in stage . [ 5 ] The Summer split 's format is split into three stages . The first stage will consist of a double round-robin group stage where all seventeen ( 17 ) teams are split into three groups of 4 and a group of 5 . This first stage will implement the " Fearless Draft " , a drafting format seen in China 's LDL developmental league where Champions from previous games in the

top four nirvana group teams advance to the single-elimination , king-of-the hill tournament . The bottom two ascend group teams and the top four nirvana group teams will have to contest an additional match . [ 6 ] As per usual , the champion for Summer 2024 will qualify for the 2024 World Championship as China 's number one seed . Spring [ edit ] Regular Season [ edit ] Spring 2024 is the second-consecutive season where the Top seed only lost one game during the entirety of the regular season . Moreover , Bilibili Gaming is the first organization

2024 LPL season Add links From Wikipedia , the free encyclopedia Sports season 2024 LPL season League LPL Sport League of Legends Duration 22 January – 20 April ( Spring ) TBD ( Summer ) Number of teams 17 Spring Split Champions Bilibili Gaming Runners-up Top Esports Season MVP Zhuo " knight " Ding ( Bilibili Gaming ) Summer Split LPL seasons ← 2023 2025 → The 2024 LPL season is the 12th and ongoing season of the League of Legends Pro League ( LPL ) , a Chinese professional esports league for the video game League of Legends .

Similar to its previous yearly splits , the 2024 LPL season will be divided into two splits : Spring and Summer . The Spring Split began on 22 January and will end on 20 April 2024 for the Grand Finals . Meanwhile , the Summer Split will begin in the latter part of 2024 following the conclusion of the Mid-Season Invitational ( MSI ) 2024 . Bilibili Gaming was crowned the 2024 Spring Champions after defeating Top Esports 3–1 in the Grand Finals rematch of the Upper Bracket Finals . This was Bilibili Gaming 's first organizational title under the

name Bilibili Gaming in the LPL . [ 1 ] The team previously defeated Top Esports in the Upper Bracket Finals 3-2 [ 2 ] and Top Esports defeating defending MSI champions JD Gaming 3–1 in the Lower Bracket Finals . [ 3 ] Bilibili Gaming is set to qualify for MSI 2024 's Group Stage alongside the champions of the LCS , LCK and LEC , while Top Esports will be pooled with the runner-up teams to qualify for the play-in tournament as per the new MSI rules . Format [ edit ] The Spring Split will have seventeen

thus far to go back-to-back 15-1 records in two different splits . BLG went 15–1 in the Summer Split and finished Spring 2024 with another 15–1 record . [ 7 ] [ 8 ] Pos Team Pld W L PCT GW GL +/- Qualification 1 Bilibili Gaming 16 15 1 0.938 30 5 +25 Advance to Upper Semifinals 2 Top Esports 16 13 3 0.813 29 8 +21 3 JD Gaming 16 13 3 0.813 27 12 +15 Advance to Round 3 4 FunPlus Phoenix 16 11 5 0.688 24 16 +8 5 Ninjas in Pyjamas 16 10 6 0.625

Gaming will return to MSI 2024 as one of the four group-stage qualifying teams . Top Esports ' defeat means their qualification to MSI 2024 will be through the Play-In stage where they are group alongside the 2023 League of Legends World Champions T1 . [ 10 ] Bracket [ edit ] Round 1 Round 2 Round 3 Upper semifinals Upper final Final 1 Bilibili Gaming 3 4 FunPlus Phoenix 1 5 Ninjas in Pyjamas 1 5 Ninjas in Pyjamas 3 5 Ninjas in Pyjamas 3 1 Bilibili Gaming 3 8 Oh My God 1 9 Team WE 2 2

April 2024 . ^ " All teams qualified for MSI 2024 League of Legends | ONE Esports " . www.oneesports.gg . 20 April 2024 . Retrieved 20 April 2024. v t e League of Legends Pro League Teams Anyone 's Legend Bilibili Gaming Edward Gaming FunPlus Phoenix Invictus Gaming JD Gaming LGD Gaming LNG Esports Ninjas in Pyjamas Oh My God Rare Atom Royal Never Give Up Team WE Top Esports TT Gaming Ultra Prime Weibo Gaming Seasons 2020 2021 2022 2023 2024 v t e 2024 in professional League of Legends competition International Mid-Season Invitational Esports World Cup World Championship Regional LCS LEC LCK LPL PCS Retrieved from " https : //en.wikipedia.org/w/inde x.php ? title=2024_LPL_season & oldid=1237480104 " Categories : League of Legends Pro League seasons 2024 in esports Hidden categories : Use dmy dates from April 2024 Articles with short description Short description matches Wikidata

**GPT-4o Summarize: 2** 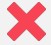

Figure 13: Response and middle results comparison of GPT-4o (OpenAI, 2024b), Qwen2-VL-7B (Qwen Team, 2024), and LLaVA-OneVision-7B (Li et al., 2024b) in the end-to-end task.

**Round3 Summarization**

**Qwen2-VL**

**Full-page Screenshot:**

**Content:**

twelfth year of China 's professional League of Legends league . The summer season sees major changes of LPL structure , including introduction of " Fearless Draft " rules and return of group stages . Seventeen teams play against each other first in four double round robins , and then in two single round robins . The top ten teams advance to the playoffs . Overview [ ] Format [ ] Two groups , Group Ascend having 9 teams and Group Nirvana having 8 Single Round Robin within group Matches are best of three The Top 7 teams in Group

in : Chinese Tournaments , Competitions , Premier Events LPL 2024 Summer < LPL | 2024 Season Sign in to edit History Talk ( 0 ) European Pro League Season 3 vs 2 September 2024 12:00:00 +0000 LIVE • PRM 2nd Div 2025 Spring Promotion vs 2 September 2024 16:00:00 +0000 LIVE • arrMY Summer League 2024 vs 2 September 2024 16:00:00 +0000 LIVE • arrMY Summer League 2024 vs 2 September 2024 17:00:00 +0000 LIVE • arrMY Summer League 2024

to Google Calendar Social Media & Links Contents 1 Overview 1.1 Format 2 Participants 2.1 Group Ascend 2.2 Group Nirvana 3 Results 4 Match Schedule 5 VODs & Match Links 6 Individual Awards 6.1 " Man of the Match " Standings 6.2 Weekly Award 6.3 Season Awards 7 Media 7.1 Streams 7.2 Broadcast Talent 7.2.1 English 7.2.2 Mandarin 7.2.2.1 Stage / Studio Hosts 7.2.2.2 Studio Hosts 7.2.2.3 Casters 7.2.2.4 Guests Casters 7.3 Additional Content 7.4 Viewership Statistics 7.5 Announcements 8 Home Venues 9 References The 2024 League of Legends Pro League ( LPL ) is the second split of the

Ascend qualify for Playoffs The Bottom 2 teams in Group Ascend and Top 4 teams in Group Nirvana qualify for Play-in Stage The Bottom 4 teams in Group Nirvana are not qualified for playoffs Show Tiebreaker Rules Hide Tiebreaker Rules If two teams have the same number of series won , ties will be broken by : Game Differential Head to Head record Participants [ ] Show Rosters Hide Rosters Rosters By Game Player Chart Group Ascend [ ] Anyone 's Legend Ale Croco Shanks Hope Kael Tabe Qingsi Bilibili Gaming Bin Xun Wei Knight Elk

Invictus Gaming YSKM glfs neny Ahn Vampire Rashomon Oh My God Hery Tianzhen Angel Starry ppgod noname Geitang Rare Atom Xiaoxu Xiaohao VicLa Assum Jwei Deceit JMZ RNG Juice Geju XBY Tangyuan Xzz huanfeng Iwandy Ming Teacherma May Team WE Wayward Yanxiang FoFo Able Mark chengz Zoom TT Gaming HOYA Beichuan ucal 1xn Feather AFei Ultra Prime Qingtian H4cker Yuekai Doggo Niket Xiaobai Yuzhang Results [ ] Group Ascend Legend Round 4 Seed Round 3 Seed Round 2 Seed Round 1 Seed Play-In Seed Team Series

vs 2 September 2024 18:15:00 +0000 LIVE • arrMY Summer League 2024 vs 2 September 2024 18:15:00 +0000 LIVE • arrMY Summer League 2024 vs 2 September 2024 19:00:00 +0000 LIVE • NACL 2025 Spring Promotion vs 2 September 2024 22:00:00 +0000 LIVE • Emerald League Special Edition vs 2 September 2024 23:00:00 +0000 LIVE • NACL 2025 Spring Promotion vs 3 September 2024 00:00:00 +0000 LIVE • LCK CL 2024 Summer Playoffs vs

1 Playoffs Split 2 Replacements 2 Split 2 Playoffs Split 3 Grand Finals 2024 Season Overview Placements Split 1 Replacements 1 Split 1 Playoffs Split 2 Replacements 2 Split 2 Playoffs Split 3 Grand Finals Overview Spring Season Spring Playoffs Summer Placements Summer Season Summer Playoffs Regional Finals Championship Points Overview Team Rosters Picks & Bans Scoreboards Runes Match History Champion Stats Player Stats LPL 2024 Summer Season Tournament Information Organizer TJ Sports Location & Dates Region CN China Event Type Offline Country China Start Date 2024-07-05 End Date 2024-07-31 Broadcast Streams Twitch Tencent Full List Schedule Spoiler-Free Schedule Export

• Emerald League Special Edition vs 5 September 2024 23:00:00 +0000 LIVE • NACL 2024 Spring Promotion TBD vs TBD 6 September 2024 00:00:00 +0000 LIVE • LCK CL 2024 Summer Playoffs vs TBD 6 September 2024 08:00:00 +0000 LIVE • European Pro League Season 3 vs 6 September 2024 10:00:00 +0000 LIVE • European Pro League Season 3 vs 6 September 2024 12:00:00 +0000 LIVE • PRM 2nd Div 2025 Spring Promotion TBD vs TBD 6 September 2024 16:00:00 +0000 LIVE • LFL 2025 Promotion

**Qwen2-VL Summarize: 9** 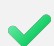

Figure 14: Response and middle results comparison of GPT-4o (OpenAI, 2024b), Qwen2-VL-7B (Qwen Team, 2024), and LLaVA-OneVision-7B (Li et al., 2024b) in the end-to-end task.

## Round3 Summarization

**LLaVA-OneVision**

**Full-page Screenshot:**

**Content:**

register and log in to edit our pages . You can also join our # leagueoflegends contributor community on our Discord . Liquipedia app match pages updated ! Liquipedia app match pages are overhauled ! Download on Android or iOS ! Liquipedia app 's match pages got completely revamped with game data , standings , VODs and more ! Download the the latest version on iOS or Android and read our update blog here . LPL 2024 Summer - Rumble Stage From Liquipedia League of Legends Wiki < LPL | 2024/Summer v Group " Ascend " ( High Group

) 2.2 Rumble Stage Group " Nirvana " ( Low Group ) 2.3 Detailed Results 2.3.1 Week 1 2.3.2 Week 2 2.3.3 Week 3 2.3.4 Week 4 2.3.5 Week 5 3 References Format [ edit ] Rumble Stage Single round robin within each group All matches are Bo3 using Standard Draft . For the Group " Ascend " ( High Group ) : Top 2 teams get seeded into the semifinals 3rd and 4th place teams get seeded into the fourth round of playoffs 5th and 6th place teams get seeded into the d e LPL & LDL/LSPL 2024 LPL Group " Ascend " ( High Group ) [ edit ] Group " Ascend " Week 5 Week 1 Week 2 Week 3 Week 4 Week 5 1 . Anyone 's Legend 1-0 2-0 +2 1 . LNG Esports 1-0 2-0 +2 1 . Weibo Gaming 1-0 2-0 +2 Bilibili Gaming 0-0 0-0 0 LGD Gaming 0-0 0-0 0 Top Esports 0-0 0-0 0 7 . FunPlus Phoenix 0-1 0-2 -2 7 . JD Gaming 0-1 0-2 -2 7 . Ninjas in Pyjamas 0-1 0-2 -2 1 . LNG Esports 3-0 6-1 +5 2 . Anyone 's Legend ▼1 3-1

2024 - 17:00 CST Match Page 36:33 22:46 MVPs : Bin , knight Bans Game 1 Game 2 JDG 0 2 TES JD Gaming 0 2 Top Esports July 31 , 2024 - 19:00 CST Match Page 41:08 26:12 MVPs : 369 , 369 Bans Game 1 Game 2 Match was held in Beijing . References [ edit ] Retrieved from " https : //liquipedia.net/leagueof legends/index.php ? title=LPL/2024/Summer/Rum ble_Stage & oldid=828570 " Hidden categories : Pages reading from original match table Pages storing into original game table Pages storing into original match table Do you want to help ? Just LNG LGD Gaming 1 2 LNG Esports July 27 , 2024 - 15:10 CST Match Page 32:40 30:05 30:02 MVPs : haichao , Hang , Weiwei Bans Game 1 Game 2 Game 3 Match was held in Suzhou . EDG 2 0 UP EDward Gaming 2 0 Ultra Prime July 27 , 2024 - 17:10 CST Match Page 29:19 34:56 MVPs : Cryin , Jiejie Bans Game 1 Game 2 FPX 0 2 TES FunPlus Phoenix 0 2 Top Esports July 27 , 2024 - 19:30 CST Match Page 33:43 24:02 MVPs :

19:10 CST Match Page 26:56 27:10 MVPs : Zika , Weiwei Bans Game 1 Game 2 Match was held in Shenzhen . Week 2 [ edit ] Week 2 July 8 , 2024 EDG 0 2 WE EDward Gaming 0 2 Team WE July 8 , 2024 - 17:10 CST Match Page 27:19 29:46 MVPs : Yanxiang , Able Bans Game 1 Game 2 Match was held in Xi'an . LGD 0 2 AL LGD Gaming 0 2 Anyone 's Legend July 8 , 2024 - 19:10 CST Match Page 28:09 28:36 MVPs , 2024 - 17:10 CST Match Page

Spring Summer Regional Finals Championship Points LDL Seeding Stage 1 Stage 2 Stage 3 Season Finals Others Demacia Cup Legend Cup LCC Click on the " Show " link on the right to see the full list 2023 LPL Spring Summer Regional Finals Championship Points LDL Stage 1 Stage 2 Stage 3 Others Demacia Cup LCC 2022 LPL Spring Summer Regional Finals Championship Points LDL Spring Summer Others Demacia Cup LCC 2021 LPL Spring Summer Regional Finals Championship Points LDL Spring Summer Others Demacia Cup LCC 2020 LPL Spring Summer Regional Finals Championship LNG Esports July 7 , 2024 -

33:22 41:37 MVPs : Xiaohao , Xiaoxu Bans Game 1 Game 2 JDG 0 2 WBG JD Gaming 0 2 Weibo Gaming July 6 , 2024 - 19:10 CST Match Page 43:33 33:25 MVPs : Tarzan , Breathe Bans Game 1 Game 2 Match was held in Beijing . July 7 , 2024 UP 0 2 iG Ultra Prime 0 2 Invictus Gaming July 7 , 2024 - 17:10 CST Match Page 31:35 39:02 MVPs : Glfs , Ahn Bans Game 1 Game 2 NIP 0 2 LNG Ninjas in Pyjamas 0 2 July 27 , 2024 LGD 1 2

**LLaVA-OneVision Summarize: 8** ❌

Figure 15: Response and middle results comparison of GPT-4o (OpenAI, 2024b), Qwen2-VL-7B (Qwen Team, 2024), and LLaVA-OneVision-7B (Li et al., 2024b) in the end-to-end task.

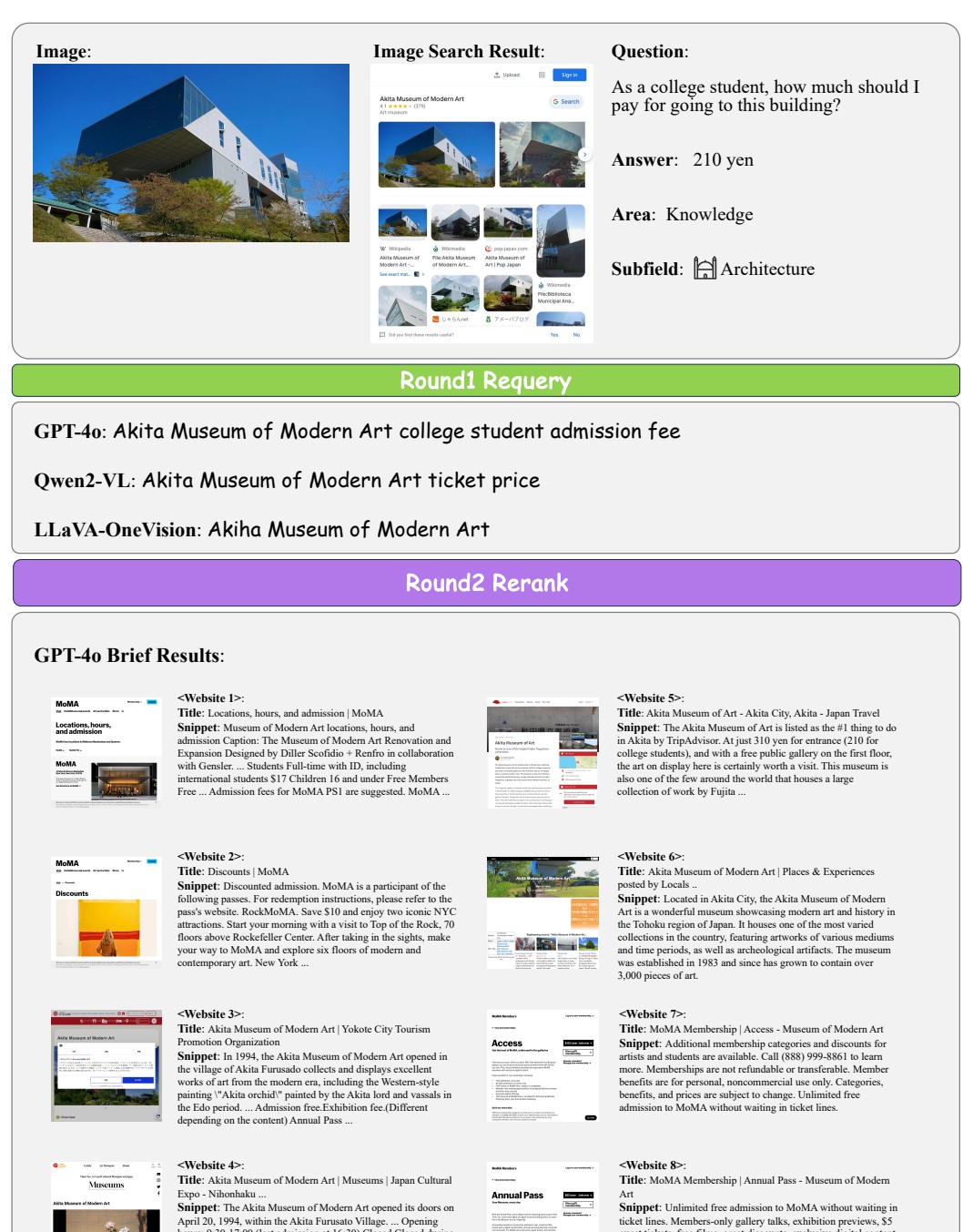

Figure 16: Response and middle results comparison of GPT-4o (OpenAI, 2024b), Qwen2-VL-7B (Qwen Team, 2024), and LLaVA-OneVision-7B (Li et al., 2024b) in the end-to-end task.

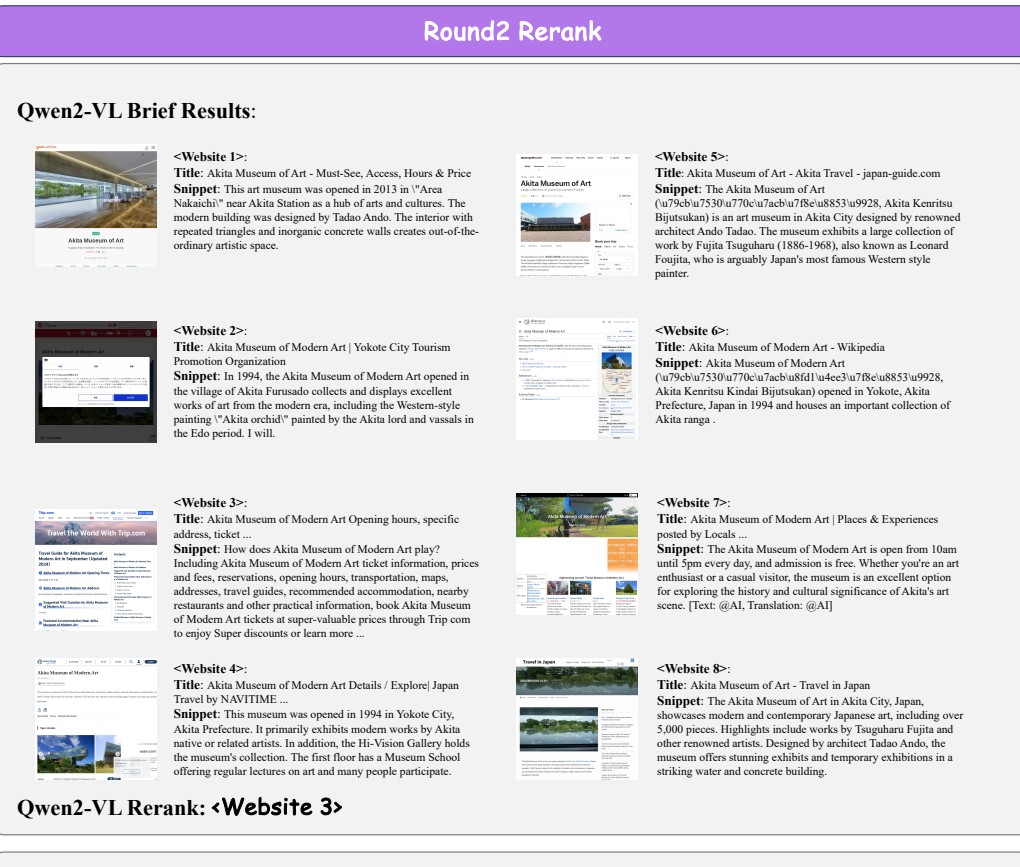

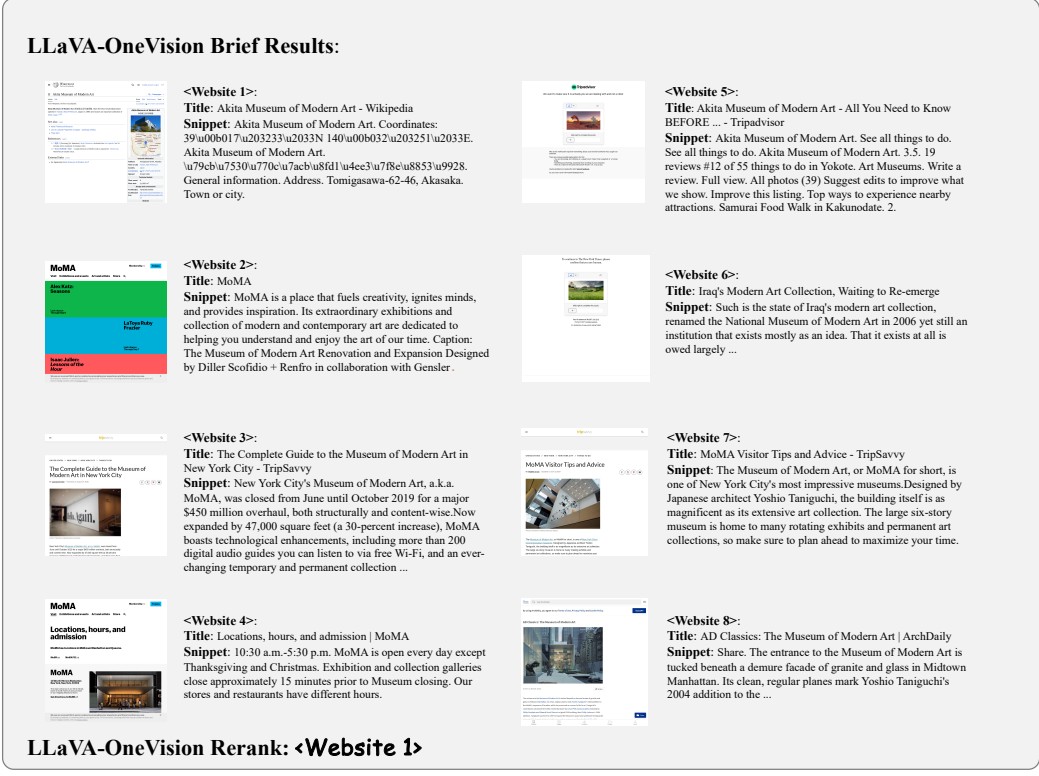

Figure 17: Response and middle results comparison of GPT-4o (OpenAI, 2024b), Qwen2-VL-7B (Qwen Team, 2024), and LLaVA-OneVision-7B (Li et al., 2024b) in the end-to-end task.

Figure 18: Response and middle results comparison of GPT-4o (OpenAI, 2024b), Qwen2-VL-7B (Qwen Team, 2024), and LLaVA-OneVision-7B (Li et al., 2024b) in the end-to-end task.

## Round3 Summarization

**Qwen2-VL**

**Full-page Screenshot:**

**Content:**

Reviews US $ 66.00 View     Top Restaurant Picks Near Akita Museum of Modern Art 1 . Bar Pasaporte Address : 204-1 Aza Takuboshita Fuke Otsutsumi Distance : 445m Bar Pasaporte No reviews yet Other Cuisine View 2 . Kuidoraku Price : $ 8.00 Address : 7-2 Ekimaecho , Yokote , Akita 013-0036 Distance : 2.28km Kuidoraku No reviews yet Bars/Bistros US $ 8.00 View 3 . Korakuen Yokoteten Price : $ 5.00 Address : It is 28-1 , Sanmaibashi in Maego , Yokote-shi , Akita character Distance : 2.03km Korakuen Yokoteten No reviews yet Other Chinese Cuisine US $

5.00 View 4 . Ganso Kamiya Yakisoba Restaurant Address : Nakano-117-67 Oyashinmachi , Yokote , Akita 013-0051 , Japan Distance : 1.17km Ganso Kamiya Yakisoba Restaurant No reviews yet Fast Food View     Verified Reviews of Akita Museum of Modern Art 第二号爱人：遇上秋田杆灯季，还是挺热热闹闹的一个节日在美术馆里 面的话，也有这种节日的气氛，氛围都还是相当不错的， 而且的话里面虽然没有什么特别多的名画名品，但是因为 是免费进入，所以值得参观。Jedy Tan：属于小众景点了，秋田县本来就不大，美术馆还在一个小 市里。但参观过能看出日本人近现代对西洋艺术的崇尚 xiaomoufa：哈哈哈，这是一个非常美丽的美术馆，特别有意思的一个 景点。E30 ＊＊＊ 67：蛮大的美术馆，里面画有些我们国家古代的味道，蛮不错 的     Also Popular With Visitors to Akita Museum of Modern Art 1 . Sendai Umino-Mori Aquarium Price : $ 14.30 Discount : $ 2.04 Recommended sightseeing time : : 4-5 hours Address : Japan , 〒983-0013

and free . everything related to the ninjas were very fun . Edo Wonderland Nikko Edomura 4.5 /5 43 Reviews No.3 of Best Things to Do in Nikko Theme Parks From US $ 37.43 View Contents Akita Museum of Modern Art Opening Times Akita Museum of Modern Art Address Suggested Visit Duration for Akita Museum of Modern Art Featured Accommodation Near Akita Museum of Modern Art 1 . Hotel Plaza Annex Yokote 2 . Yokote Central Hotel 3 . Quad Inn Yokote 4 . Yokote Plaza Hotel Top Restaurant Picks Near Akita Museum of Modern Art 1 . Bar Pasaporte

Hotels Flights Trains Cars Car Rentals Airport Transfers App Customer Support USD Search Bookings Sign in / Register Travel the World With Trip.com Travel Guide for Akita Museum of Modern Art in September ( Updated 2024 )     Akita Museum of Modern Art Opening Times Year round : 9:30-17:00     Akita Museum of Modern Art Address 62-46 , Akasaka Tomigazawa | Inside Akita Furusato Mura , Yokote , Akita Prefecture     Suggested Visit Duration for Akita Museum of Modern Art 1-2 hours Featured Accommodation Near Akita Museum of Modern Art 1 . Hotel Plaza Annex Yokote Address 2 . Kuidoraku 3 . Korakuen Yokoteten 4 . Ganso Kamiya Yakisoba Restaurant Verified Reviews of Akita Museum of Modern Art Also Popular With Visitors to Akita Museum of Modern Art 1 . Sendai Umino-Mori Aquarium 2 . Tsugaru-han Neputa mura Village 3 . Suntopia World 4 . Edo Wonderland Nikko Edomura Contents Akita Museum of Modern Art Opening Times Akita Museum of Modern Art Address Suggested Visit Duration for Akita Museum of Modern Art Featured Accommodation Near Akita Museum of Modern Art 1 . Hotel Plaza Annex Yokote 2 . Quad Inn Yokote 3 . Quad Inn Yokote 4 . Yokote Plaza Hotel Top Restaurant Picks Near Akita Museum of Modern Art 1 . Bar Pasaporte 2 . Kuidoraku 3 . Korakuen Yokoteten 4 . Ganso Kamiya Yakisoba Restaurant Verified Reviews of Akita Museum of Modern Art Also Popular With Visitors to Akita Museum of Modern Art 1 . Sendai Umino-Mori Aquarium 2 . Tsugaru-han Neputa mura Village 3 . Suntopia World 4 . Edo Wonderland Nikko Edomura Popular Travelogues Bangkok Travelogue | Manila Travelogue | Tokyo Travelogue | Taipei Travelogue | Hong Kong Travelogue | Seoul Travelogue | Kuala Lumpur Travelogue | Los Angeles Travelogue | Shanghai Travelogue

$ 14.30 View 2 . Tsugaru-han Neputa mura Village Price : $ 3.38 Address : Japan , 〒036-8332 Aomori , Hirosaki , Kamenokomachi , 6 1 Distance : 0.63 mi Tsugaru-han Neputa mura Village No reviews yet From US $ 3.38 View 3 . Suntopia World Price : $ 8.85 Recommended sightseeing time : : 0.5-1 day Address : 1-1 Kubo , Agano , Niigata 959-2212 , Japan Distance : 5.06 mi What travelers say : M515shunyi1618 : Very suitable for family outings of playground , Ferris wheel , pirate ship are fun . Suntopia World 5 /5 9 Reviews Amusement

Ekimaechou Price : $ 53.00 Distance : 2.23km Hotel Plaza Annex Yokote 4.3 /5 37 Reviews -5 % US $ 53.00 View 2 . Yokote Central Hotel Address : Heiwacho 9-10 Price : $ 50.00 Distance : 2.91km Yokote Central Hotel 3.7 /5 18 Reviews -5 % US $ 50.00 View 3 . Quad Inn Yokote Address : Sekibata-52-1 Yasuda Price : $ 58.00 Distance : 1.9km Quad Inn Yokote 4.1 /5 14 Reviews US $ 58.00 View 4 . Yokote Plaza Hotel Address : 7-1 Ekimaecho Price : $ 66.00 Distance : 2.27km Yokote Plaza Hotel 3.9 /5 10

**Qwen2-VL Summarize: US $ 66.00** 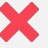

Figure 19: Response and middle results comparison of GPT-4o (OpenAI, 2024b), Qwen2-VL-7B (Qwen Team, 2024), and LLaVA-OneVision-7B (Li et al., 2024b) in the end-to-end task.

Figure 20: Response and middle results comparison of GPT-4o (OpenAI, 2024b), Qwen2-VL-7B (Qwen Team, 2024), and LLaVA-OneVision-7B (Li et al., 2024b) in the end-to-end task.

