SUPPLEMENTARY MATERIAL OVERVIEW

- Section A: Related work.
- Section B: Automated data curation pipeline.
- Section C: Additional dataset details.
- Section D: Future direction.
- Section E: Additional experiments and analysis.
- Section F: Additional experimental details.
- Section G: More dataset details.
- Section H: Qualitative examples.

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

**Question**: In the LPL 2024 Summer season, how many teams were in Group Ascend?

**Answer**: 9

**Area**: News

**Subfield**: ⊗ Sports

### Round1 Requery

**GPT-4o**: LPL 2024 Summer season Group Ascend teams count

**Qwen2-VL**: LPL 2024 Summer season Group Ascend teams

**LLaVA-OneVision**: LPL 2024 Summer season Group Ascend teams

### Round2 Rerank

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

Akita Museum of Modern Art مصرى日本語 Edit links Coordinates : 39°17′33″N 140°32′51″E / 39.29250°N 140.54750°E / 39.29250 ; 140.54750 From Wikipedia , the free encyclopedia Building in Akita Prefecture , Japan Akita Museum of Modern Art 秋田県立近代美術館 Wikimedia | © OpenStreetMap General information Address Tomigasawa-62-46 , Akasaka Town or city Yokote , Akita Prefecture Country Japan Coordinates 39°17′33″N 140°32′51″E / 39.29250°N 140.54750°E / 39.29250 ; 140.54750 Opened 20 April 1994 Technical details Floor count 8 Floor area 11,166.5 m 2 Design and construction Architect ( s ) Yamashita Sekkei Architecture firm http : //www.yamashitasekkei.co . jp/en/works/list/momaakit
a.html Website homepage ( ja ) Akita Museum of Modern Art ( 秋田県立近代美術館 , Akita Kenritsu Kindai Bijutsukan ) opened in Yokote , Akita Prefecture , Japan in 1994 and houses an important collection of Akita ranga . [ 1 ] [ 2 ] See also [ edit ] Wikimedia Commons has media related to Akita Museum of Modern Art . Akita Prefectural Museum List of Cultural Properties of Japan - paintings ( Akita ) Yōga ( art ) References [ edit ] ^ 概要 [ Summary ] ( in Japanese ) . Akita Prefecture . Archived from the original
on 19 October 2013 . Retrieved 19 October 2013 . ^ 県立近代美術館（横手） [ Akita Museum of Modern Art ] ( in Japanese ) . Yokote City . Retrieved 19 October 2013 . External links [ edit ] ( in Japanese ) Akita Museum of Modern Art hide Authority control databases International VIAF National United States Japan Academics CiNii People ISIL : JP-2000539 This article related to a museum in Japan is a stub . You can help Wikipedia by expanding it . v t e Retrieved from " https : //en.wikipedia.org/w/inde x.php ? title=Akita_Museum_of_Mod ern_Art & oldid=1199934740 " Categories :
Yokote , Akita Art museums and galleries in Akita Prefecture Prefectural museums Art museums and galleries established in 1994 1994 establishments in Japan Important Cultural Properties of Akita Prefecture Japanese museum stubs Hidden categories : Pages using gadget WikiMiniAtlas CS1 uses Japanese-language script ( ja ) CS1 Japanese-language sources ( ja ) Use dmy dates from November 2019 Articles with short description Short description is different from Wikidata Infobox mapframe without OSM relation ID on Wikidata Coordinates on Wikidata Articles containing Japanese-language text Commons category link is on Wikidata Articles with Japanese-language sources ( ja ) All stub articles Pages using the Kartographer extension

**LLaVA-OneVision Summarize:** free ❌

Figure 13: Response and middle results comparison of GPT-4o (OpenAI, 2024b), Qwen2-VL-7B (Qwen Team, 2024), and LLaVA-OneVision-7B (Li et al., 2024a) in the end-to-end task.