# OpenReview forum: "MMSearch: Unveiling the Potential of Large Models as Multi-modal Search Engines"
_ICLR.cc/2025/Conference — ICLR 2025 Poster_

### Official Review · Reviewer_gUra · 2024-10-30

**Soundness:** 3
**Presentation:** 3
**Contribution:** 2
**Rating:** 6
**Confidence:** 3

**Summary:**

The authors present MMSEARCH-ENGINE, a multimodal AI search engine pipeline, and MMSEARCH, a comprehensive evaluation benchmark to assess the multimodal search performance of LMMs.

**Strengths:**

1. This paper introduces the first mm RAG pipeline (at least based on the related work introduced) and the first mm search evaluation.
2. The methods for building the MM RAG pipeline and evaluation are solid and with comprehensive statistics.
3. The methodology statement is easy to follow.
4. Comprehensive experiments.
5. Step-wise evaluation of the RAG pipeline is cool.

**Weaknesses:**

1. The evaluation data creation pipeline is not efficient, which may result in difficulties to keep the evaluation dynamic.
2. Evaluation data only comprises 300 entries, which is far from comprehensive. Generally, a benchmark should have 1k+ samples to be robust.
3. The authors didn't exhibit the step-wise scores for the proposed RAG pipeline, which harms the result soundness.

**Questions:**

NA

---

> ### Author Response · Authors · 2024-11-22
>
> We sincerely appreciate your valuable comments. We found them extremely helpful in improving our manuscript. We address each comment in detail, one by one below.
>
> > **Comment 1: Inefficient data creation pipeline**
>
> Thank you for your advice. In response to your concern, we have developed an automated data curation pipeline. The introduction of this efficient pipeline largely simplify the data annotation process. The data creation pipeline reduces the time of the steps of human browsing the news website and raising the questions, which consumes most of the time when we are collecting data of MMSearch. For future data evolution, all we have to do is to specify the time range for the collected questions and let the pipeline do the work. Only minimal human effort is required to guarantee the quality of the benchmark. We hope this automatic data creation pipeline could address your concern of the difficulty in keeping the benchmark and evaluation dynamic.  Please refer to the **general response**s for our proposed automated methods to further expand the benchmark.
>
> > **Comment 2: Benchmark size is limited**
>
> Thank you for your advice. Here are **a few reasons why the current scale is enough**:
>
> 1. The news domain in MMSearch covers all hot topics available on the mainstream news website like CNN or The Guardian. It includes most daily search scenarios with comprehensive categories of news.
> 2. Both the two domain data in MMSearch aim to be beyond the internal knowledge of LMMs. Different from other benchmarks, all the data in MMSearch truly examines the searching capability without any leakage.
> 3. Each instance includes four tasks: end-to-end, requery, rerank, and summarization. A model is required to be evaluated on each task with the task-specific annotated data. Therefore, there are 1,200 questions in total in MMSearch.
>
> The current scale also allows for efficient evaluation. In the end-to-end setting, we need to interact with the search engine and also record the search results and screenshots. The evaluation of 300 questions now takes 5 to 8 hours for the end-to-end task to complete. A larger dataset may harm the development efficiency.
>
> In addition, we have developed an automated data curation pipeline. During the rebuttal phase, ***we have collected 300 more data*** from the CNN and The Guardian websites following our designed pipeline.  All these data are about news that occured between September 1, 2024 to November 14, 2024, ensuring no overlap with the current benchmark. We believe the automated pipeline could be employed to further expand the dataset at low cost.
>
> >  **Comment 3: Step-wise scores missing**
>
> Sorry for any confusion caused. We have included the step-wise scores for the proposed pipeline in Table 2 in the main paper during submission. We list the scores of the three steps: requery, rerank, and summarizaion. ***We have refined the table caption in Table 2 to make the content of the table more clear.***

---

> > ### Author Response · Authors · 2024-11-30
> >
> > Dear Reviewer gUra,
> >
> > We recognize that the timing of this discussion period may not align perfectly with your schedule, yet we would greatly value the opportunity to continue our dialogue before the deadline approaches.
> >
> > Could you let us know if your concerns have been adequately addressed? If not, please feel free to raise them, and we are more than willing to provide further clarification; if you find that your concerns have been resolved, we would highly appreciate it if you could reconsider the review score.
> >
> > We hope that we have resolved all your questions, but please let us know if there is anything more.
> >
> > Thanks.

---

> > > ### Author Response · Authors · 2024-12-03
> > >
> > > Dear Reviewer gUra,
> > >
> > > We recognize that the timing of this discussion period may not align perfectly with your schedule, yet we would greatly value the opportunity to continue our dialogue before the deadline approaches.
> > >
> > > Could you let us know if your concerns have been adequately addressed? If not, please feel free to raise them, and we are more than willing to provide further clarification; if you find that your concerns have been resolved, we would highly appreciate it if you could reconsider the review score.
> > >
> > > We hope that we have resolved all your questions, but please let us know if there is anything more.
> > >
> > > Thanks.

---

### Official Review · Reviewer_mwVo · 2024-11-02

**Soundness:** 3
**Presentation:** 3
**Contribution:** 3
**Rating:** 6
**Confidence:** 5

**Summary:**

This paper introduces MMSearch, a benchmark that evaluates the capacity of large multimodal models (LMMs) as AI-driven search engines for handling complex multimodal queries (text and image). The study proposes MMSearch-Engine, a tailored pipeline enabling LMMs to address multimodal search tasks, divided into stages: requery, rerank, and summarization. Experimental results show that models like GPT-4o, integrated into this pipeline, perform better than current commercial systems, such as Perplexity Pro, in end-to-end multimodal search tasks. The paper presents detailed error analyses, offering insights into the limitations of LMMs in search-oriented subtasks, especially in requery and rerank.

**Strengths:**

(1) This paper constructs a novel benchmark MMSearch. It fills an important gap in multimodal AI evaluation by focusing on search capabilities and interaction with multimodal data. It offers a unique benchmark that pushes beyond standard image-text alignment tasks.

(2) This paper proposes an effective multimodal retrieval pipeline MMSearch-Engine, which consists of requery, rerank, and summarization. Experiments have shown that this pipeline can effectively improve the performance of the model.

(3) The experimental analysis is comprehensive and provides good reference conclusions. The error analysis and comparison between commercial and open-source models are valuable for understanding current model limitations, providing useful insights for improving multimodal search models.

(4) This paper uses a weighted score of four scores to evaluate the effect of the model, which not only focuses on the correctness of the model's results, but also focuses on the correctness of its process, which can provide a more comprehensive and effective understanding of the model's performance.

**Weaknesses:**

(1) Although MMSearch covers a wide range of news and knowledge domains, the total number is only 300 instances. Such a number and scale are not enough to fully reflect the generality of the model, etc. The author may need to further expand the scale of the dataset in the future.

(2) Lack of task complexity hierarchy. The complexity of current MMSearch tasks is relatively consistent, lacking a hierarchy of tasks from simple to complex. In the future, tasks of different difficulty levels can be designed to better measure the performance of the model when dealing with tasks of increasing complexity.

(3) There are still limitations in the validation of the model's adaptability. The adaptability of this method in different fields and application scenarios is still unclear, especially on data from special or professional fields (such as medicine or law). Introducing datasets from these fields can more comprehensively evaluate the versatility and adaptability of the model.

**Questions:**

(1) In future versions, could the pipeline include user feedback to improve task accuracy iteratively?

(2) How general and adaptable is the approach to data from specific or specialized fields, such as medicine or law?

---

> ### Author Response · Authors · 2024-11-22
> **Official Comment by Authors (1/3)**
>
> > **Comment 1: Further expand the scale of the dataset**
>
> Thank you for your advice. Here are ***a few reasons why the current scale may be enough***:
>
> 1. The news domain in MMSearch covers all hot topics available on the mainstream news website like CNN or The Guardian. It includes most daily search scenarios with comprehensive categories of news.
> 2. Both the two domain data in MMSearch aim to be beyond the internal knowledge of LMMs. Different from other benchmarks, all the data in MMSearch truly examines the searching capability without any leakage.
> 3. Each instance includes four tasks: end-to-end, requery, rerank, and summarization. A model is required to be evaluated on each task with the task-specific annotated data. Therefore, there are 1,200 questions in total in MMSearch.
>
> The current scale also allows for efficient evaluation. In the end-to-end setting, we need to interact with the search engine and also record the search results and screenshots. The evaluation of 300 questions now takes 5 to 8 hours for the end-to-end task to complete. A larger dataset may harm the development efficiency.
>
> In addition, we have developed an automated data curation pipeline. During the rebuttal phase, ***we have collected 300 more data*** from the CNN and The Guardian websites following our designed pipeline.  All these data are about news that occured between September 1, 2024 to November 14, 2024, ensuring no overlap with the current benchmark. We believe the automated pipeline could be employed to further expand the dataset at low cost.
>
> Please refer to the **general responses** for more details of our proposed automated data curation pipeline.

---

> ### Author Response · Authors · 2024-11-22
> **Official Comment by Authors (2/3)**
>
> > **Comment 2: Lack of complexity hierarchy**
>
> Thank you for pointing out! We have implemented a three-tier complexity assessment framework based on the key challenges in the three steps of MMSearch-Engine:
>
> 1. **Requery difficulty.** This concerns the complexity of transforming the original question into an effective search query. Complex cases arise when the question references image content, requiring the LMM to first analyze the visual information and then synthesize it with the text question into a coherent search query. For instance, when a user asks about a landmark shown in an image, the LMM must first identify the landmark through image search and then incorporate this information into a text query about the landmark's specific attributes.
> 2. **Rerank difficulty.** This dimension evaluates the challenge of identifying and prioritizing relevant search results. The difficulty primarily scales with the information scarcity. If there are only very limited websites containing useful information, it is more difficult to successfully retrieve and choose the website.
> 3. **Summarization difficulty.** This aspect involves both information synthesis and multi-modal reasoning challenges. In cases requiring synthesis, answers cannot be derived from a single source sentence - the LMM must integrate information scattered across different parts of the website. For example, comparing event frequencies across locations (like concert counts between cities) requires gathering and analyzing distributed data. Additionally, some questions demand analysis of both textual and visual website content, sometimes necessitating comparison with input images with images in the website.
>
> Based on these criteria, we have categorized all questions into three difficulty levels, with the following distribution: hard (28%), medium (27.7%), and easy (44.3%). We showcase the evaluation results grouped by the difficulty levels as below:
>
> | Model                                       | All  |        |      | End-to-end |        |      | Requery |        |      | Rerank |        |      | Summarization |        |      |
> | - | - | - | - | - | - | - | - | - | - | - | - | - | - | - | - |
> |                                             | Easy | Middle | Hard | Easy       | Middle | Hard | Easy    | Middle | Hard | Easy   | Middle | Hard | Easy          | Middle | Hard |
> | **Closed-source LLMs with MMSEARCH-ENGINE** |      |        |      |            |        |      |         |        |      |        |        |      |               |        |      |
> | GPT-4o (OpenAI, 2024b)                      | 63.1 | 63.9   | 59.2 | 61.0       | 62.3   | 57.4 | 54.3    | 41.2   | 40.6 | 82.2   | 85.6   | 81.5 | 64.0          | 65.5   | 59.0 |
> | **Open-source LLMs with MMSEARCH-ENGINE**   |      |        |      |            |        |      |         |        |      |        |        |      |               |        |      |
> | LLaVA-OneVision (Li et al., 2024b)          | 42.9 | 32.3   | 31.1 | 36.8       | 24.6   | 23.3 | 46.5    | 28.9   | 25.8 | 78.0   | 67.8   | 69.8 | 51.5          | 56.7   | 53.6 |
> | Qwen2-VL_AnyRes (Qwen Team, 2024)           | 45.6 | 44.9   | 45.3 | 41.2       | 38.9   | 40.4 | 46.5    | 34.0   | 32.2 | 73.5   | 83.3   | 74.7 | 50.3          | 56.8   | 59.6 |
> | LLaVA-OneVision (72B)                       | 52.2 | 48.0   | 49.0 | 47.6       | 41.6   | 44.0 | 51.0    | 39.6   | 33.2 | 79.5   | 85.1   | 83.3 | 60.3          | 63.7   | 60.6 |
> | Qwen2-VL_AnyRes (72B)                       | 53.7 | 53.4   | 50.2 | 50.7       | 48.6   | 46.9 | 52.3    | 40.0   | 37.1 | 72.3   | 82.2   | 77.8 | 58.5          | 67.0   | 53.6 |
>
> The result showcases that most LMMs still have trouble handling the hard questions compared with the easy questions. The complete version is in **Table 4** in Appendix C. We have included the discussion in **Appendix C** in the revised version. These complexity annotations will be included in our benchmark release to facilitate future research on model capabilities across different difficulty levels.

---

> > ### Author Response · Authors · 2024-11-23
> > **Official Comment by Authors (3/3)**
> >
> > > **Comment 3 & 5: The method's adaptability to professional fields such as medicine and law.**
> >
> > Thank you for your advice. For the medical field, we test our method on two benchmarks: the knowledge-based split in SLAKE [1] and Medmcqa [2]. As for the law field, we test on Legalbench [3]. We compute the F1 score for all the benchmark. We use GPT-4o as the base LMM for evaluation. Due to the constrained time of the rebuttal phase, we report 50 randomly sampled data for each field for the end-to-end task evaluation. The result is listed below:
> >
> > | Model  | Medical | Law  |
> > | ------ | ------- | ---- |
> > | GPT-4o | 53.4    | 88.3 |
> >
> > The F1 scores of these professional fields are close to or even surpass the F1 score in the general domain data in MMSearch benchmark, demonstrating our method's excellent adaptability to various fields.
> >
> > > **Comment 4: Include user feedback to improve task accuracy iteratively**
> >
> > We appreciate this insightful suggestion regarding user feedback integration. Our pipeline can indeed be enhanced through interactive user feedback loops. When the model produces an incorrect answer, users can identify the specific step where the error occurred and prompt the model to reconsider its reasoning. This iterative process allows for guided model refinement until accurate results are achieved.
> > We conducted preliminary experiments on GPT-4o with this approach on three test cases. The results demonstrated that the LMM successfully interpreted user feedback and appropriately adjusted its responses based on the provided guidance. The model's ability to understand the user input and revise its reasoning suggests significant potential for improving task accuracy.
> > We have included the discussion for future research direction in **Appendix D** of our revised version.

---

> ### Comment · Reviewer_mwVo · 2024-11-23
>
> Thank you for the detailed response to my questions, it addressed most of my concerns. I have a further suggestion regarding the validation of the proposed datasets, particularly given the limited number of instances. Conducting robustness experiments, comparing the results with human evaluations, and performing significance testing could help strengthen the findings.

---

> ### Author Response · Authors · 2024-11-25
>
> Thank you for your suggestion!
>
> 1. We conduct the ***robustness analysis*** comprising iterative experiments (n=8) utilizing LLaVA-OneVision-7B with a temperature parameter of 0.6 across three primary tasks: requery, rerank, and summarization. The results, visualized in **Fig. 9 in Appendix E.2**, demonstrate remarkable stability across all tasks, with minimal variance in performance metrics, suggesting the robustness of the evaluation results.
>
> 2. As for the ***significance testing***, we evaluate the correlation between the ***human evaluation*** and the automatic evaluation metrics of the requery and summarization tasks. Since the rerank task can be evaluated quantitatively as a classification problem using human annotations as ground truth, we focused our human evaluation on the other two tasks. Three independent annotators assess the quality of the requery and summarization outputs using a score of {1, 2, 3, 4}. Then the score is normalized to a [0,1] range. We compute two correlation coefficients: Pearson's correlation coefficient ($r$) and Spearman's rank correlation coefficient ($\rho$) for the significance testing. The results are:
>
>    | Task    | Pearson's r | Spearman's ρ |
>    | ------------- | ----------- | ------------ |
>    | Requery       | 0.5413      | 0.5503       |
>    | Summarization | 0.8112      | 0.7944       |
>
>    The analysis revealed substantial correlations between human and automatic evaluations across both tasks. All correlations were highly significant (p $\textless$ 0.001), providing strong evidence for the validity of our automatic evaluation metrics in aligning with human judgment.

---

> > ### Author Response · Authors · 2024-11-30
> >
> > Dear Reviewer mwVo,
> >
> > We recognize that the timing of this discussion period may not align perfectly with your schedule, yet we would greatly value the opportunity to continue our dialogue before the deadline approaches.
> >
> > Could you let us know if your concerns have been adequately addressed? If not, please feel free to raise them, and we are more than willing to provide further clarification; if you find that your concerns have been resolved, we would highly appreciate it if you could reconsider the review score.
> >
> > We hope that we have resolved all your questions, but please let us know if there is anything more.
> >
> > Thanks.

---

> ### Author Response · Authors · 2024-12-03
>
> Dear Reviewer mwVo,
>
> We recognize that the timing of this discussion period may not align perfectly with your schedule, yet we would greatly value the opportunity to continue our dialogue before the deadline approaches.
>
> Could you let us know if your concerns have been adequately addressed? If not, please feel free to raise them, and we are more than willing to provide further clarification; if you find that your concerns have been resolved, we would highly appreciate it if you could reconsider the review score.
>
> We hope that we have resolved all your questions, but please let us know if there is anything more.
>
> Thanks.

---

### Official Review · Reviewer_3MNQ · 2024-11-03

**Soundness:** 3
**Presentation:** 4
**Contribution:** 3
**Rating:** 8
**Confidence:** 4

**Summary:**

This paper is about benchmarking LLMs as multimodal search engines. The authors propose:
1) A pipeline that enables this type of multimodal search, where images are translated into textual queries before querying a search engine and summarizing the results.
2) A small, but high-quality dataset as benchmark. They do their best to ensure that there is no information leakage between the LLM's training data and the benchmark, by collecting long-tail knowledge data, and very recent news.
They present results using commercial search engines, closed-source, and open-source LLMS, and show that there is much room for improvement, analysing the sources of error along the way.

**Strengths:**

1) The paper is very well-written, with excellent presentation and clarity. In my view, the scope is very well-defined, and there is no redundant information.
2) I particularly like that the authors try to create a dataset that LLMs have not seen before.
3) There is a very large appendix with interesting additional experiments and qualitative analysis.

**Weaknesses:**

1) The way the authors construct the dataset is not future-proof. We have to assume that every piece of news that comes out is immediately ingested, at the very least by large corporations with commercial search engines. In that sense, the benchmark will be obsolete in a couple of months. I would love to see a piece of discussion that proposes a way to make the data collection and annotation pipeline future-proof.
2) The benchmark dataset, even though high quality, is rather small. Maybe I am missing something, but are there any indications that its coverage over the problem space is sufficient?

**Questions:**

Please, refer to the "Weaknesses" section.

I have another question for the authors that I do not consider a weakness, just interesting: In line 409 you state "Any-resolution input only provides slight or no improvement.". Have you tried to corrupt a clean image with differend kinds of corruptions (e.g., occlusions, blur, gaussian noise, etc.), and see how it impacts the search results?

---

> ### Author Response · Authors · 2024-11-22
>
> We sincerely appreciate your valuable comments. We found them extremely helpful in improving our manuscript. We address each comment in detail, one by one below.
>
> >  **Comment 1: The future-proof issue of the benchmark**
>
> Thank you for your advice. In response to the concern, we have developed an automatic data curation pipeline. The pipeline automatically obtains latest news from mainstram news websites efficiently. As for the question and annotation, we employ a model pool containing state-of-the-art LMMs to raise the questions and provide the answers. To guarantee the quality of the generated data, we further conduct human verification. The high efficiency brought by the data generation pipeline enables us to update the benchmark in a low cost. Therefore, the regular update of the benchmark could prevent it from being obsolete by substituing the benchmark with recent news.
>
> Please refer to the **general responses and Appendix B** for more details of our proposed data curation pipeline.
>
> > **Comment 2: Indication of the sufficiency of the benchmark's coverage**
>
> Thank you for pointing this out. ***The categories of our benchmark are based on the categories of the authority news website like CNN [1] and The Guardian [2].*** We assume the categories listed in these popular websites could cover most search scenarios.
>
> Specifically, take CNN as example, the header categories of the website are listed as follow: *US, World, Politics, Business, Health, Entertainment, Style, Travel, Sports, Science, Climate, Weather, and Wars*. We omit politically sensitive categories (Politics, Wars) and related news from US and World sections to maintain neutrality. We merge Style, Travel, Health, Climate, and Weather into a single "General" category, as these topics typically involve less time-sensitive information. We preserve Business, Entertainment, Sports, and Science as standalone categories and add a "False Premise" category to test model robustness. The knowledge domain categories are derived from these news categories and validated by confirming that even state-of-the-art LMMs have limited prior knowledge in these areas, ensuring the necessity of external information retrieval.
>
> > **Comment 3: The impact of corruption on the input image**
>
> This is a very interesting question. It helps to find out the role of image input in the search process. We consider four kinds of image corruptions: occlusion, gaussian blur, gaussian noise, and color jitter. The corruption is added to all the images. To better understand where the image input affects most, we conduct the experiments respectively on the requery, rerank, and summarization tasks. We test on InternVL2 for both low-resolution and any-resolution settings. The results are shown below:
>
> | Resolution | Corruption     | Requery | Rerank | Summarization |
> | ---------- | -------------- | ------- | ------ | ------------- |
> | Low        | -              | 32.3    | 46.5   | 48.5          |
> | Low        | occlusion      | 31.8    | 58.7   | 46.9          |
> | Low        | gaussian blur  | 30.7    | 58.5   | 47.3          |
> | Low        | gaussian noise | 29.9    | 57.2   | 45.2          |
> | Any        | -              | 31.4    | 53.2   | 46.9          |
> | Any        | occlusion      | 29.3    | 57.0   | 46.7          |
> | Any        | gaussian blur  | 31.7    | 50.7   | 45.8          |
> | Any        | gaussian noise | 30.0    | 53.5   | 45.4          |
>
> The observations are as follows:
>
> 1. The requery and summarization step highly rely on the image input. When applying image corruption, most results are worse than clear image input.
> 2. The image input is not important for the rerank step, even bringing interference. The image corruption even brings large improvement to the low-resolution setting, which suggests InternVL2 does not leverage image information in the rerank step.
> 3. Any-resolution image input is more robust to image corruption than low-resolution image input. Compared to the low-resolution setting, the any-resolution setting suffers less performance deterioration.
>
> [1] https://edition.cnn.com/
>
> [2] https://www.theguardian.com/international

---

> > ### Comment · Reviewer_3MNQ · 2024-11-25
> >
> > Dear authors, thank you for providing additional analysis with image corruptions, and for addressing my questions and concerns.

---

### Official Review · Reviewer_7DLE · 2024-11-04

**Soundness:** 4
**Presentation:** 4
**Contribution:** 4
**Rating:** 6
**Confidence:** 4

**Summary:**

This work describes a new pipeline to empower LMMs with multimodal search capabilities called MMSearch-Engine, and introduce MMSearch, a comprehensive evaluation benchmark to assess the multimodal search performance of LMMs which includes 300 high-quality instances across 14 subfields. The study also points out LMMs' challenges in multimodal search and suggests scaling test-time computation could improve AI search engines.

**Strengths:**

* MMSearch-Engine is a well-designed three steps pipeline that empower LLMs with multimodal search capabilities.
* MMSearch is a fair evaluation benchmark because its data is all dated later than the model's knowledge base update time.
* Case analysis is comprehensive.

**Weaknesses:**

* The small amount of data may lead to a certain degree of randomness.
* When constructing the data pipeline, human annotators perform multiple requery operations in cases where no website is classified as valid. Only one requery operation may not fully demonstrate the LMMs' capabilities.

**Questions:**

* Due to the sequential nature of requery, rerank, and summarization, errors in each step can lead to subsequent errors. How to ensure the reasonableness of the weight of each score in formula 1?

---

> ### Author Response · Authors · 2024-11-22
>
> We sincerely appreciate your valuable comments. We found them extremely helpful in improving our manuscript. We address each comment in detail, one by one below.
>
> >  **Comment 1: Small amount of data may cause randomness**
>
> Thank you for your advice. Here are ***a few reasons why the current scale may be enough***:
>
> 1. The news domain in MMSearch covers all hot topics available on the mainstream news websites like CNN or The Guardian. It includes most daily search scenarios with comprehensive categories of news.
> 2. Both the two domain data in MMSearch aim to be beyond the internal knowledge of LMMs. Different from other benchmarks, all the data in MMSearch truly examines the searching capability without any leakage.
> 3. Each instance includes four tasks: end-to-end, requery, rerank, and summarization. A model is required to be evaluated on each task with the task-specific annotated data. Therefore, there are 1,200 questions in total in MMSearch.
>
> The current scale also allows for efficient evaluation. In the end-to-end setting, we need to interact with the search engine and also record the search results and screenshots. The evaluation of 300 questions now takes 5 to 8 hours for the end-to-end task to complete. A larger dataset may harm the development efficiency.
>
> In addition, we have developed an automated data curation pipeline. During the rebuttal phase, ***we have collected 300 more data*** from the CNN and The Guardian websites following our designed pipeline.  All these data are about news that occured between September 1, 2024 to November 14, 2024, ensuring no overlap with the current benchmark. We believe the automated pipeline could be employed to further expand the dataset at low cost.
>
> Please refer to the general responses for more details of our proposed automated data curation pipeline.
>
> > **Comment 2: One requery operation may not demonstrate the LMMs' capabilities.**
>
> 1. Sorry for the confusion caused. In the data annotation phase, we want to make sure at least one search result contains the ground truth answer to the question. So we require annotators to repeatedly adjust their requery to satisfy the condition. In contrast, in the evaluation phase, for fairness, each LMM is given only one chance to provide a good requery. This standardized approach allows for consistent evaluation metrics.
>
> 2. We acknowledge the valuable suggestion to explore multiple requery attempts.  This could be another evaluation setting where we could better exploit LMM's capability. Following this suggestion, we conducted an additional experiment with LLaVA-OneVision (7B), where we generated 5 requery attempts per question (temperature = 1.0) and selected the best requery based on similarity to human-annotated queries. The result is provided below.
>
>
> | Model       | Req. Times | Avg  | News | Know. |
> | ----------- | ---------- | ---- | ---- | ----- |
> | LLaVA-OV-7B | 1          | 29.6 | 33.1 | 19.7  |
> | LLaVA-OV-7B | 5          | 30.1 | 33.3 | 21.3  |
>
> The result showcases that conducting requery for five times could indeed improve the end-to-end performance. This is a valuable insight for future development of a multimodal AI search engine pipeline, and we have included this discussion in our revised manuscript, highlighted in **Appendix E.1**.
>
> > **Comment 3: The reasonableness of the score weight given the sequential nature of each step**
>
> Thanks for pointing out. We would like to explain the rationality of our weighting scheme from two complementary perspectives:
>
> 1. ***The importance of each task due to the sequential nature is already reflected in the end-to-end score.*** Detailedly, although it is easy to discern that the upstreamed task is more important, it is difficult to assign a precise weight to each of them. So we do not manually assign the weights but directly focus mainly on the end-to-end score, which implicitly considers their cascaded nature.
> 2. ***The individual task weights serve as complementary metrics rather than indicators of relative importance.*** Relying solely on end-to-end evaluation, while comprehensive, may obscure the performance characteristics of individual components and hinder targeted improvements. We, therefore, maintain independent evaluation of each task, with the weight distribution designed to balance the prominence of the end-to-end metric against the component-level assessments. This dual evaluation strategy enables both system-level optimization and component-specific refinements.
>
> We have included this discussion in **Appendix F** in our revised version.

---

> > ### Author Response · Authors · 2024-11-30
> >
> > Dear Reviewer 7DLE,
> >
> > We recognize that the timing of this discussion period may not align perfectly with your schedule, yet we would greatly value the opportunity to continue our dialogue before the deadline approaches.
> >
> > Could you let us know if your concerns have been adequately addressed? If not, please feel free to raise them, and we are more than willing to provide further clarification; if you find that your concerns have been resolved, we would highly appreciate it if you could reconsider the review score.
> >
> > We hope that we have resolved all your questions, but please let us know if there is anything more.
> >
> > Thanks.

---

> > > ### Author Response · Authors · 2024-12-03
> > >
> > > Dear Reviewer 7DLE,
> > >
> > > We recognize that the timing of this discussion period may not align perfectly with your schedule, yet we would greatly value the opportunity to continue our dialogue before the deadline approaches.
> > >
> > > Could you let us know if your concerns have been adequately addressed? If not, please feel free to raise them, and we are more than willing to provide further clarification; if you find that your concerns have been resolved, we would highly appreciate it if you could reconsider the review score.
> > >
> > > We hope that we have resolved all your questions, but please let us know if there is anything more.
> > >
> > > Thanks.

---

### Author Response · Authors · 2024-11-22
**Response To All Reviewers (1/2)**

We thank all reviewers for their insightful comments and great questions. We greatly appreciate all the reviewers' **acknowledgment of our paper's motivation, design details, and experimental analysis**. Here, we are sharing a summary response covering the common concerns of reviewers.

**Benchmark Size and Future Expansion**

Reviewers commented about the relatively small benchmark size, which is 300 questions in total. We present several observations that may support the appropriateness of the benchmark scale. Then, we propose an automated/semi-automated data curation pipeline to expand the benchmark efficiently. The introduction of the pipeline could not only address the concern of limited data scale but also enable high efficiency for benchmark evolution.

Here are **a few reasons why the current scale is enough**:

1. The news domain in MMSearch covers all hot topics available on mainstream news websites like CNN or The Guardian. It includes most daily search scenarios with comprehensive categories of news.
2. Both the two domain data in MMSearch aim to be beyond the internal knowledge of LMMs. Different from other benchmarks, all the data in MMSearch truly examines the searching capability without any leakage.
3. Each instance includes four tasks: end-to-end, requery, rerank, and summarization. A model is required to be evaluated on each task with the task-specific annotated data. Therefore, there are 1,200 questions in total in MMSearch.

The current scale also allows for efficient evaluation. In the end-to-end setting, we need to interact with the search engine and also record the search results and screenshots. The evaluation of 300 questions now takes 5 to 8 hours for the end-to-end task to complete. A larger dataset may harm the development efficiency.

Now we introduce our **automated/semi-automated data curation pipeline**. The figure is shown in **Figure 8 in Appendix B**. We focus on the news domain here. We first define a website pool and a model pool. The website pool contains general news websites like CNN and expertise websites like Arxiv. The model pool contains state-of-the-art models for the data curation pipeline to guarantee diversity and fairness.

1. End-to-end data curation.  We employ a web crawler to obtain all the subsites published later than a specific date. However, not all the websites are suitable for raising a question to test the LMMs' searching capability. For example, some websites do not contain any recent news, while some websites' contents are difficult to convert into a question with a definite answer. Therefore, we randomly choose a model from the model pool to serve as a news filter, prompting it to filter valid websites by providing some few-shot examples. Next, we provide the text contents and screenshots of the valid websites to a model from the model pool. The model is asked to raise several questions based on the website content. It is encouraged to raise questions that cannot be answered only by text. As for the question with an image, the model is asked to describe the image content briefly, and we will later use the description to search in Bing and obtain the first result image. Finally, we apply the quality check of the generated questions and their corresponding images either by human or a model from the model pool.
2. Requery data generation. We provide the question, the question image content, and the answer to a model. The model is prompted to come up with a suitable requery.
3. Rerank data generation. We provide the generated requery to the search engine and retrieve K websites for rerank. Again, we provide the question, the question image content, and the answer to a model and ask to categorize each website into valid, unsure, or invalid.
4. Summarization data generation. We randomly choose one website from the websites marked as valid in the last step and obtain its full content.

Notably, human check is a must for the requery data generation process. There should be at least one valid website to guarantee the effectiveness of the generated requery. Only after human verification of this step, the quality of rerank and summarization data generation is assured.

---

> ### Author Response · Authors · 2024-11-22
> **Response To All Reviewers (2/2)**
>
> During the rebuttal phase, **we have collected 300 more data** from the CNN and The Guardian websites following our designed pipeline.  All these data are about news that occurred between September 1, 2024 to November 14, 2024, ensuring no overlap with the current benchmark. Due to the limited time, we employ the LMM to conduct the question check step. Our manual inspection indicates the generated data meets acceptable quality standards.  For the revised manuscript, we will conduct manual verification of all the newly generated end-to-end data and complete the requery, rerank and summarization data generation process.
>
> We evaluate LLaVA-OneVision-7B, Qwen2-VL-7B, and GPT-4o on the newly collected data on the end-to-end task. The results are:
>
> | Model       | Avg  | Sports | Science | Entertainment | Finance | General |
> | ----------- | ---- | ------ | ------- | ------------- | ------- | ------- |
> | LLaVA-OV-7B | 40.2 | 46.0   | 33.1    | 45.3          | 56.6    | 37.0    |
> | Qwen2-VL-7B | 44.2 | 47.9   | 33.3    | 61.9          | 30.0    | 42.3    |
> | GPT-4o      | 51.7 | 51.8   | 49.4    | 76.4          | 55.2    | 48.4    |
>
> The trend of the model performance aligns with that reported in the main paper, which suggests the high quality of the generated data. The introduction of this automated pipeline could expand the dataset with much less human effort and greatly reduce the cost of data annotation. The future direction is to further refine the process and boost the data quality. We have added the content of this data curation pipeline in **Appendix B** in the revised version.

---

### Meta-Review · Area_Chair_BzrN · 2024-12-19

**Metareview:**

**Summary:**

This paper proposes a straightforward approach to developing a multi-modal search engine by integrating LLMs, Google Lens, and other tools. The core components of the system are re-querying, reranking, and summarization. The scope of the paper is well-defined and offers valuable insights into the proposed direction. Notably, the benchmark datasets provided are expected to be highly beneficial and could stimulate further research in this area, despite their relatively small size. Furthermore, the paper includes extensive experiments and detailed explanations in the Appendix, thoroughly showcasing the authors' ideas, methodology, and evaluation process.

**Strength:**
- A well-designed three-step pipeline (re-query, rerank, summarization) that enhances LLMs with multimodal search capabilities.
- Fills a critical gap in multimodal AI evaluation by emphasizing search and interaction with multimodal data, beyond standard image-text alignment tasks.
- Authors created a dataset that LLMs have not encountered before, making it a unique and valuable resource for research.

**Weakness:**
- The dataset contains only 300 instances, which is insufficient to fully reflect the generality of the model's performance.
- Human annotators perform multiple requery operations when no website is classified as valid, but a single requery may not fully demonstrate the capabilities of LLMs.
- No clear indication that the dataset sufficiently covers the problem space, raising concerns about the robustness of the benchmark.

**Additional Comments On Reviewer Discussion:**

During the discussion, the authors provided detailed responses to all reviewers; however, only two out of the four reviewers participated in the discussion. I think that the reviewers were overall positive about the paper. So, I recommend acceptance to this paper.

---

### Decision · Program_Chairs · 2025-01-22

Accept (Poster)